# Effects of management practices on the ecosystem-service multifunctionality of temperate grasslands

Franziska J. Richter [1] ✉, Matthias Suter[2], Andreas Lüscher [2], Nina Buchmann [1], Nadja El Benni[3], Rafaela Feola Conz [4], Martin Hartmann [4], Pierrick Jan[5] & Valentin H. Klaus [1,2]

Human wellbeing depends on ecosystem services, highlighting the need for improving the ecosystem-service multifunctionality of food and feed production systems. We study Swiss agricultural grasslands to assess how employing and combining three widespread aspects of grassland management and their interactions can enhance 22 plot-level ecosystem service indicators, as well as ecosystem-service multifunctionality. The three management aspects we assess are i) organic production system, ii) an eco-scheme prescribing extensive management (without fertilization), and iii) harvest type (pasture vs. meadow). While organic production system and interactions between the three management aspects play a minor role, the main effects of eco-scheme and harvest type considerably shape single services. Moreover, the eco-scheme 'extensive management' and the harvest type 'pasture' enhance plot-scale ecosystem-service multifunctionality, mostly through facilitating cultural services at the expense of provisioning services. These changes in ecosystem-service supply occur mainly via changes in land-use intensity, i.e., reduced fertilizer input and harvest frequency. In conclusion, diversifying grassland management where this is currently homogeneous across farms and landscapes depicts an important first step to improve landscape-scale multifunctionality for sustainable grassland systems. To meet societal ecosystem services demand, the three studied management aspects can be systematically combined to increase ecosystem services that are in short supply.

Providing sustainably produced food and feed while safeguarding ecosystem services is a primary global challenge[1,2]. Environmental sustainability issues of intensive production therefore set a spotlight on land management strategies to increase beneficial ecosystem services and reduce negative environmental impacts (disservices) of agriculture[3].

Before introducing agricultural policies that promote certain farming practices, the effectiveness of these practices needs to be assessed in terms of their individual and combined environmental benefits.

Grasslands are a major global land use, covering 70% of the global agricultural area[4]. Globally, grasslands are highly important for

[1]Grassland Sciences, Institute of Agricultural Sciences, ETH Zürich, Zürich, Switzerland. [2]Forage Production and Grassland Systems, Agroscope, Zürich, Switzerland. [3]Sustainability Assessment and Agricultural Management, Agroscope, Ettenhausen, Switzerland. [4]Sustainable Agroecosystems, Institute of Agricultural Sciences, ETH Zürich, Zürich, Switzerland. [5]Managerial Economics in Agriculture, Agroscope, Ettenhausen, Switzerland. ✉e-mail: fraju.richter@t-online.de

nutrition security as 20% of protein for human nutrition is derived from ruminants, which are to a large part fed from grasslands[5–7], more than half of which (by area) are located on marginal land that cannot be used for crop production[8]. Further, grasslands contribute to human well-being by providing many ecosystem services other than food or feed provisioning, including supporting and regulating ecosystem services, such as water and climate regulation, as well as cultural ecosystem services by, for instance, contributing to visually pleasing landscapes[9,10]. Preserving and promoting the ability of grasslands to provide many ecosystem services in one area, i.e., high ecosystem-service multifunctionality[11], will be crucial to support human well-being in a world faced with growing human population, urban growth, and climate change.

In many temperate regions, grasslands and their ecosystem services rely on either regular grazing by animals or mowing, as they would otherwise be encroached by shrubs and trees. However, widespread intensification of agricultural management in the form of increased fertilization as well as more frequent and earlier harvests has become a threat for grassland ecosystem-service multifunctionality, by heavily focusing on provisioning ecosystem services and neglecting other ecosystem services[12–14]. Agri-environmental strategies and policies therefore aim at enhancing ecosystem-service multifunctionality by regulating grassland management intensity, potentially resulting in losses in agricultural production.

Many of these agricultural and agri-environmental regulations target plot-scale grassland management, resulting in different plant communities and delivering different sets of ecosystem services and different levels of plot-scale multifunctionality[12]. The latter corresponds to ecosystem multifunctionality[11] and informs about how a broad set of ecosystem services is affected by a specific management practice. Knowledge on plot-scale effects of management on ecosystem services is further required to achieve multifunctionality on the landscape scale, resolving inevitable trade-off between ecosystem services at the plot scale[15,16]. However, given the many potentially interacting aspects that shape agricultural grassland management (i.e., mowing versus grazing, different fertilization levels, etc.), it has not yet been investigated how management intensity in concert with other key aspects of grassland management affect a broader range of ecosystem services and associated multifunctionality.

Here, we tested three aspects of grassland management that are widespread in their adoption and implemented independently from but alongside each other for their ability to increase ecosystem-service multifunctionality. We analyzed the impact of (i) organic production, (ii) the eco-scheme "extensive management", and (iii) the harvest type, i.e., the option to either use the land as pasture (grazing predominant) or as meadow to feed the grass offsite (mowing predominant), on 22 ecosystem-service indicators and resulting plot-scale multifunctionality. While organic management and eco-scheme "extensive management" are instruments of agri-environmental policies that financially compensate farmers for restricting management intensity, the harvest type is usually set by farmers according to their individual farming approach on the given land.

Organic management receives a lot of attention, for instance in the Farm to Fork strategy of the Common Agricultural Policy of the European Union[17], as it depicts a farm-level production system (hereafter "Production system organic versus non-organic") that minimizes synthetic inputs to promote healthy soils and ecosystems[18]. However, organic management has never been tested for its ability to enhance multiple ecosystem services in grassland ecosystems[19]. For arable crops, organic management has been found to benefit ecosystem-service multifunctionality, while reducing yields by 5–35%[20,21].

Many European countries provide economic incentives for extensive grassland management (hereafter "Eco-scheme extensive management": yes versus no), with the aim of enhancing biodiversity and potentially also specific ecosystem services such as water

quality[22–24]. Existing studies on the effect of grassland eco-schemes focused mainly on biodiversity while the impact of these schemes on other ecosystem services has not been as thoroughly investigated so far. Recent studies found several regulating, supporting, and cultural ecosystem services to be decreased at high-management intensity, indicating the importance of extensive management for non-production ecosystem-service multifunctionality[12,13]. Yet, studies assessing how the effect of extensive management on the simultaneous supply of multiple ecosystem services interacts with, e.g., the harvest type, are needed to understanding trade-offs and synergies in ecosystem services provision linked to this widespread policy tool[13].

A further key aspect of grassland management concerns the predominant type of biomass removal or harvest (hereafter "Harvest type pasture versus meadow"). The Harvest type can shape grasslands, as grazing animals are selective for or against certain plant species[25], which can lead to increasing abundance of unwanted species, while mowing is unselective and impacts all species equally. Meadows and pastures thus show distinct differences in vegetation composition but also microbial processes, which likely affects ecosystem services[26,27]. In addition, trampling by livestock can lead to disservices such as erosion and low soil organic carbon as a consequence of sward damage[28]. However, despite its ubiquitous relevance, the effect of Harvest type on ecosystem-service multifunctionality is currently unknown.

To address the question of how these three widespread management aspects, as well as the interactions among them, influence grassland ecosystem services and related multifunctionality, we assessed 22 ecosystem-service indicators in 86 managed grasslands in the Canton of Solothurn, Switzerland (Supplementary Table S1). These 22 indicators correspond to 12 ecosystem services, following the common international classification of ecosystem services (CICES[29]; Fig. 1a), as in some cases several indicators reflect different components of one ecosystem services[11,30]. Our study design allowed us to investigate all possible combinations of the management aspects of interest, namely Production system (organic vs. non-organic), Eco-scheme (extensive management yes vs. no) and Harvest type (pasture vs. meadow) on grassland plots (Fig. 1b).

Our objectives were to first analyze the effect of management aspects on single ecosystem-service indicators using multivariate regression. Second, we assessed how the effects of the three management aspects on single ecosystem services act via changes in mowing frequency, fertilizer amount and grazing intensity, the three most decisive management actions in Central European grasslands[31]. Third, we tested the effect of the three management aspects on plot-scale ecosystem-service multifunctionality by using a log response ratio approach. These consecutive analytical steps allowed us to assess if and how Production system, Eco-scheme and Harvest type affect grassland ecosystem-service multifunctionality. Results of this study can, thus, inform and support improving grassland management and related agri-environmental policies in optimizing grassland ecosystem-service provision and thus improving the multifunctionality of agricultural landscapes. Insights into the relationships between single practices and ecosystem services allow farmers and other decision-makers to adapt grassland management to support a specific ecosystem service or even ecosystem-service multifunctionality at a given site.

## Results

### Impact of management aspects on ecosystem-service indicators
The 22 ecosystem-service indicators hardly differed between Production systems (organic vs. non-organic), but often between extensive Eco-scheme vs. non-Eco-scheme grasslands, and between the two Harvest types (meadow vs. pasture; Fig. 2). For instance, biomass yield and digestibility were higher in non-Eco-scheme grasslands, while Eco-scheme grasslands performed better regarding less nitrogen (N) leaching and less surface P, our measure to

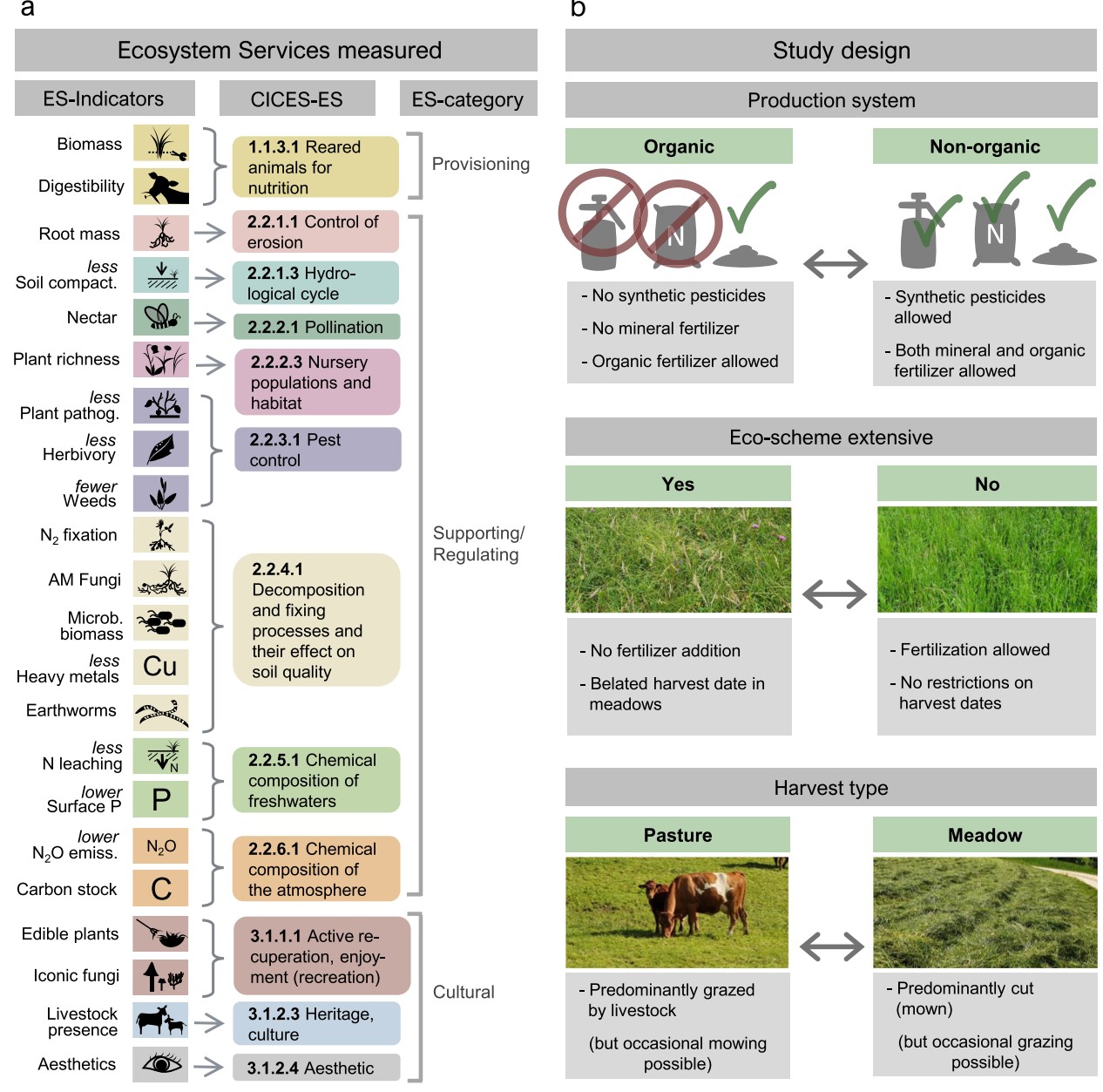

**Fig. 1 | Overview of measured ecosystem-service indicators (ES-indicators) and study design. a** From left to right: indicators grouped according to the corresponding ecosystem service defined by CICES[29] and corresponding ecosystem-service category. **b** From top to bottom: brief definition of the three management aspects studied: Production system, Eco-scheme, and Harvest type. All 22 ecosystem-service indicators were measured for the eight possible combinations of Production system, Eco-scheme, and Harvest type. Total number of study plots was 86 (see Supplementary Table S1 for number of plots per combination of management aspects).

assess eutrophication risk. Eco-scheme pastures performed especially well regarding edible plant abundance, iconic fungi, and livestock presence (Fig. 2). The main effects and interactions of the three management aspects on the 22 ecosystem-service indicators were evaluated with generalized linear latent variable models (GLLVM[32]), which revealed that Eco-scheme had the strongest influence on the ecosystem-service indicators, while interactions between the management aspects were generally not relevant to explain these indicators (Fig. 3 and Supplementary Table S2). Eco-scheme extensive management significantly improved ten out of 22 ecosystem-service indicators belonging to supporting/regulating and cultural ecosystem services, such as plant richness, proportion of AM fungi, and esthetics (Fig. 3). In comparison, management without Eco-scheme promoted six out of 22 ecosystem-service indicators, including both provisioning and some supporting/regulating indicators, such as earthworms and fewer weeds. Harvest type had a smaller influence on ecosystem-service indicators than the Eco-scheme. Use as pasture promoted five indicators (e.g., digestibility and edible plants), while use as meadow promoted five indicators (e.g., biomass yield and lower $N_2O$ emissions). Production system significantly affected only two ecosystem-service indicators: Organic management increased the relative abundance of AM fungi and led to less nitrate leaching compared to non-organic management. Yet, no ecosystem-service indicator significantly decreased (worsened) as a response to organic management (Fig. 3). Taken together, while Eco-scheme and Harvest type affected many of the ecosystem-service indicators, the

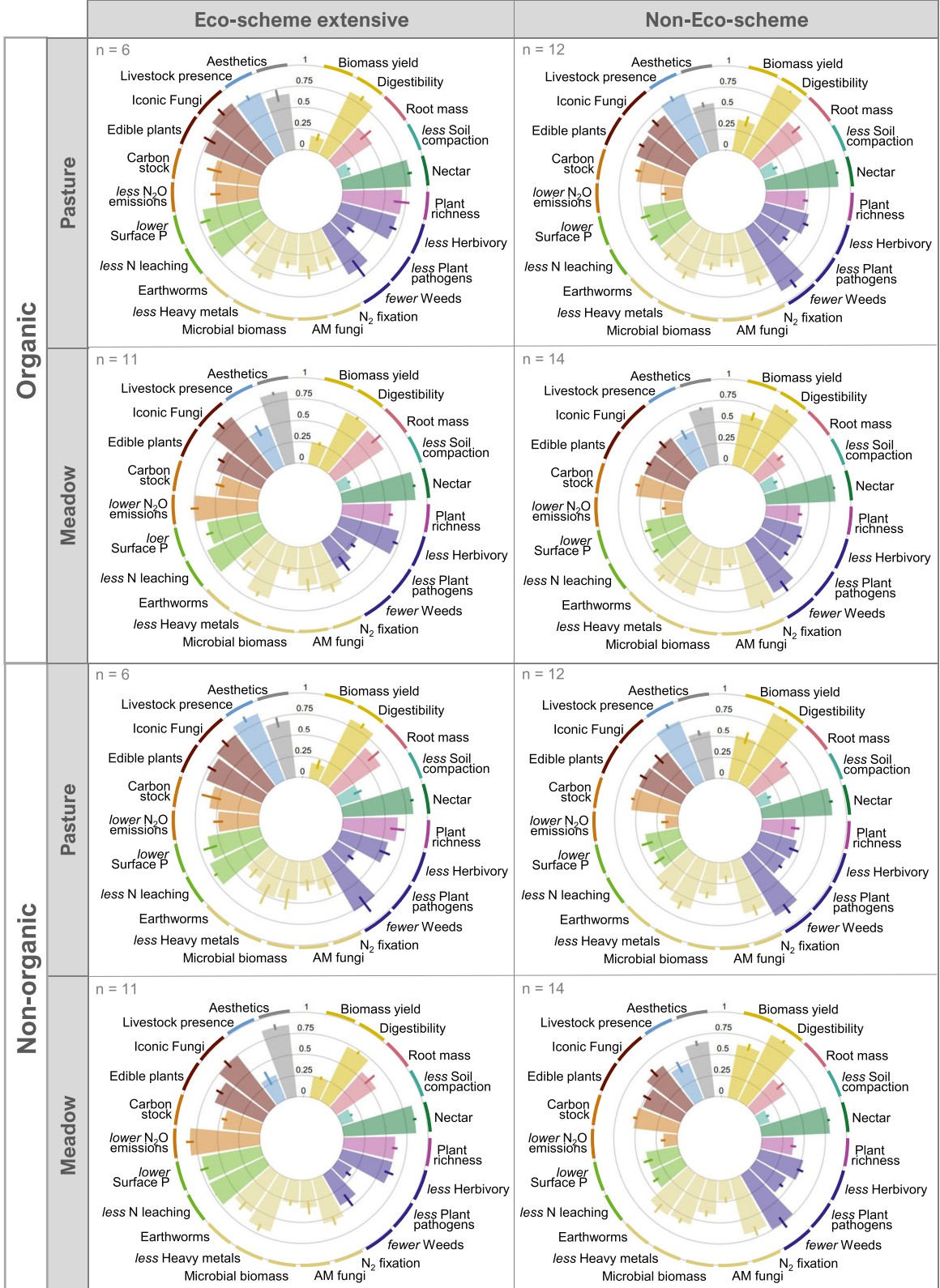

**Fig. 2 | Ecosystem-service indicators in response to three grassland management aspects studied.** Bars denote the mean value (with standard error) for each indicator and the combination of management aspects (Production system: organic vs. non-organic; Eco-scheme extensive management: yes vs. no; Harvest type: pasture vs. meadow). Values are maximum-scaled per indicator (over the whole dataset) and reversed for disservices; colors correspond to the ecosystem services according to CICES (see Fig. 1). Replicate numbers (grasslands) are given in the top left corner of the respective barplot. See Fig. 3 for statistical tests on management effects on all indicators. Source data are provided as a Source Data file.

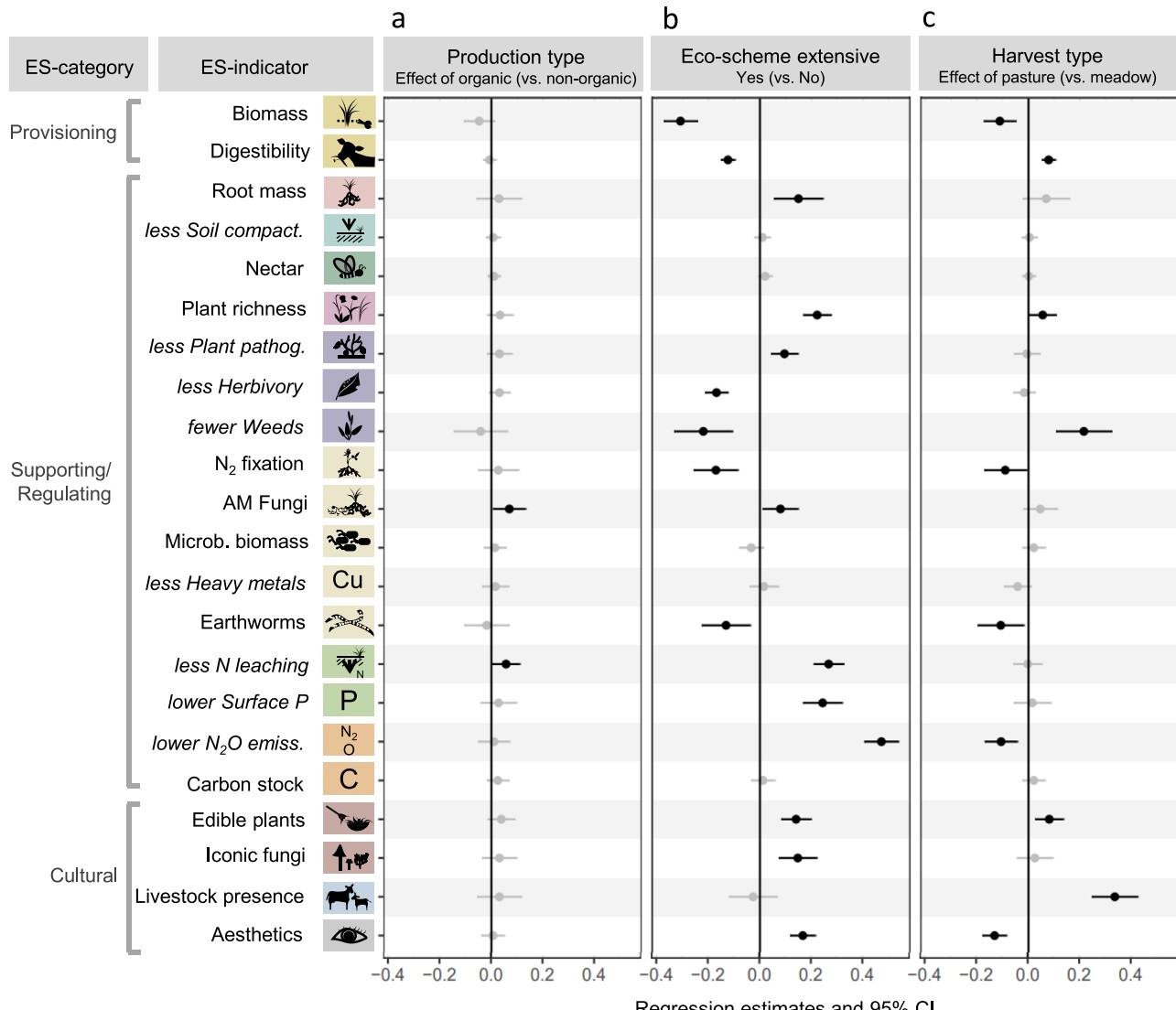

**Fig. 3 | Effects of grassland management on ecosystem-service indicators (ES-indicators) according to a generalized linear latent variable model.** Significant effects are shown in black ($P < 0.05$). Regression estimates (points) and 95% confidence intervals (bars) derived from testing the main effects of Production system (**a**), Eco-scheme (**b**), and Harvest type (**c**) on the 22 max-scaled ecosystem-service indicators. Color coding of icons for ecosystem-service indicators corresponds to the respective ecosystem service according to CICES (see Fig. 1). Ecosystem-service indicators in italics have been reversed to show services instead of disservices. This model included three environmental variables, soil pH, sand content, and elevation, the coefficient plots of which can be found in Supplementary Fig. S2. $N = 86$ grasslands. Source data are provided as a Source Data file.

Production system had only a marginal influence. This observation was consistent across the other two management aspects (no significant interactions between the Production system and Eco-scheme and Harvest type). The finally selected GLLVM also included three environmental co-variables to account for potential confounding of the environment with the management aspects (see Supplementary Table S2 for the model selection summary). Here, soil pH, sand content, and elevation affecting six, seven, and eight ecosystem-service indicators, respectively (see Supplementary Fig. S2 for the coefficient plots, and Supplementary Fig. S3 for the correlations among all tested environmental variables). Noteworthy, although the environmental variables explained some variation in the data, their addition to the GLLVM had only a marginal impact on the effects of the management aspects compared with a model without environmental variables (see Supplementary Fig. S4). This is because the management aspects were the main drivers of plot-scale responses and well distributed across the environmental gradient of the study region.

### Effects of management aspects as explained by management intensity variables

To explore the extent to which the 17 statistically significant effects of the three management aspects (observed in the GLLVM, Fig. 3) on the ecosystem-service indicators could be explained by the management intensity variables fertilizer amount, mowing frequency, and grazing intensity, we used standardized structural equation models (SEMs, Fig. 4). The basic structure of the SEM used for all 17 ecosystem-service indicators is shown in Fig. 4a (and full SEMs with model fit statistics in Supplementary Fig. S5). Note that inclination of the grasslands was significantly related to Eco-scheme and pasture (Fig. 4a), while there was no such effect of inclination on Production system because the sampling design assured pairs of organic and non-organic grasslands to have similar topography. Elevation and northness were also tested but removed from the SEM because they did not significantly affect any of the management aspects.

Eco-scheme extensive and Harvest-type pasture showed rather strong effects, decreasing grassland land-use intensity with the

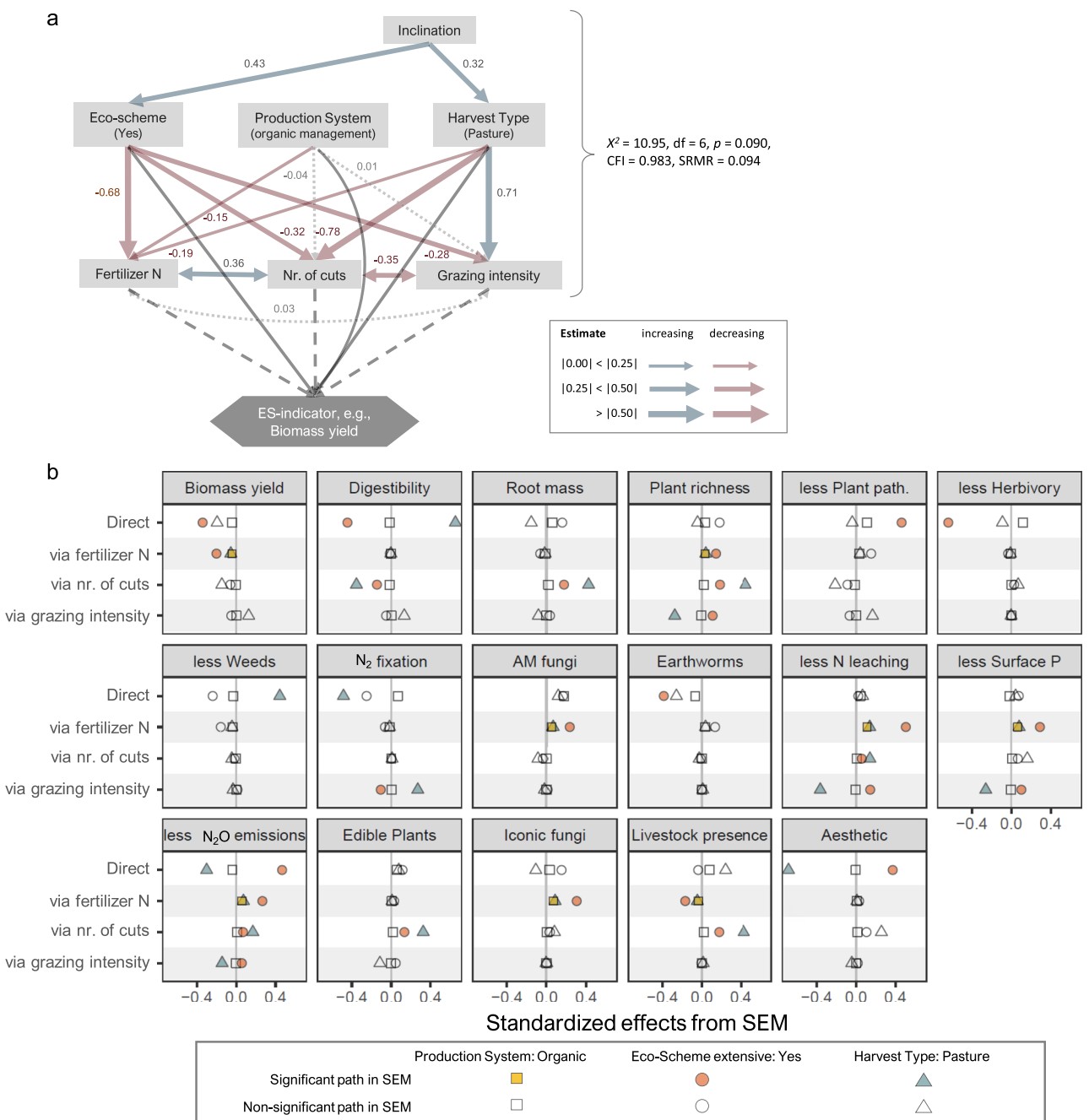

**Fig. 4 | Structural equation models (SEMs) identifying direct and indirect effects of management aspects on ecosystem-service indicators (ES-indicators).** Indirect effects of the Production system, Eco-scheme, and Harvest type act via fertilizer N (fertilization intensity), number of cuts (cutting frequency), and grazing intensity (livestock units × grazing days) on the max-scaled ecosystem-service indicators. Tested only for the 17 ecosystem-service indicators significantly influenced by at least one of the three management aspects (via GLLVM see Fig. 3). **a** This basic SEM model was used for every ecosystem-service indicator, with red arrows denoting decreasing, blue-gray arrows increasing effects, and light gray dotted arrows insignificant effects (*P* > 0.05). $\chi^2$ statistic, comparative fit index (CFI) and standardized root mean-squared residual (SRMR) of the basic model are given. Dark gray solid arrows in the lower part of the SEM show direct and dashed arrows indirect effects on the ecosystem-service indicators via fertilizer N, number of cuts, and grazing intensity. These direct and indirect effects are shown in (**b**) with filled symbols, indicating the size of direct and indirect effects from a significant path, and non-filled symbols from insignificant paths. Besides inclination, elevation was also included in the initial SEM but was removed because it did neither significantly affect the three management aspects nor the three measures of management intensity. See appendix for full SEMs and fit indices (Supplementary Fig. S5). Units for the management intensity variables are fertilizer N: plant-available N in kg ha$^{-1}$ year$^{-1}$, number of cuts: cuts year$^{-1}$, and grazing intensity: livestock unit days ha$^{-1}$ year$^{-1}$. *N* = 86 grasslands. Source data are provided as a Source Data file.

obvious exception of the Harvest-type pasture that increased grazing intensity on the cost of cutting frequency. Organic management only decreased fertilization but not mowing and grazing intensities (Fig. 4a). Organic pastures and meadows received on average

significantly less available N via fertilization (46.3 ± 52.4 kg ha$^{-1}$, 82.6 ± 42.8 kg ha$^{-1}$, mean ± SD) than non-organic pastures and meadows (75.7 ± 55.8 kg ha$^{-1}$, 111.6 ± 55.7 kg ha$^{-1}$; Fig. 4a and Supplementary Table S1). Of the N from fertilizer on non-organic grasslands, only

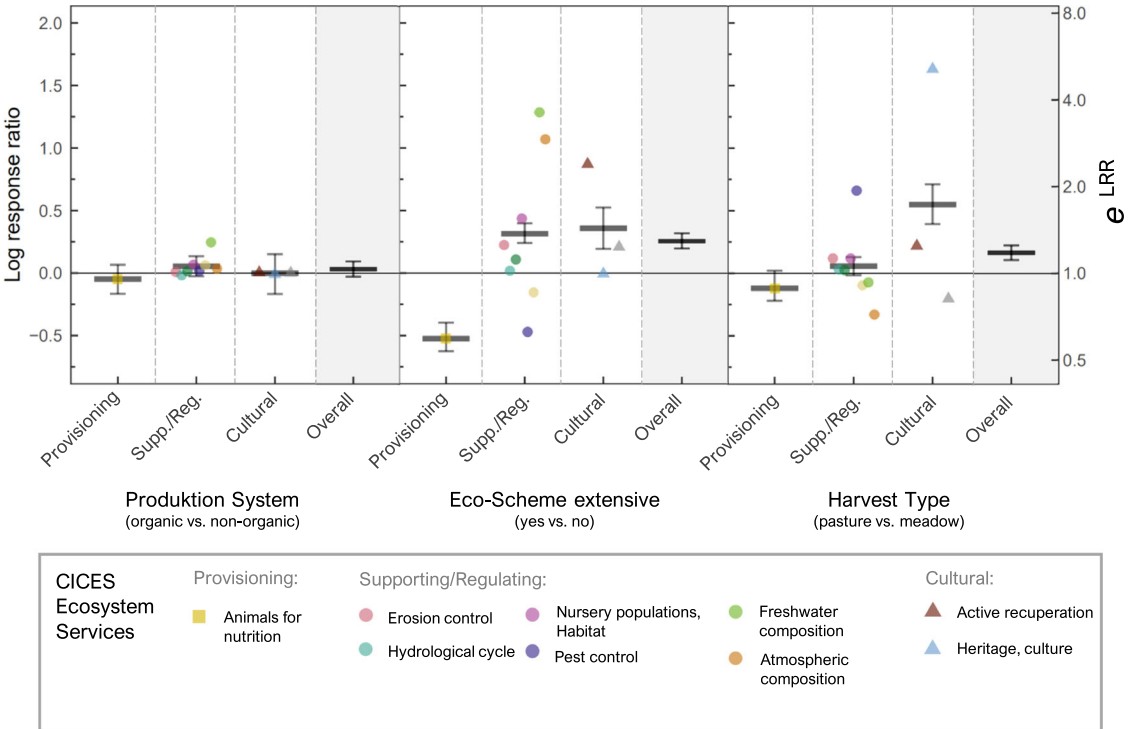

**Fig. 5 | Effects of grassland management on the multifunctionality of 12 ecosystem services, derived by aggregating log response ratios (LRRs, black horizontal bars) of 22 ecosystem-service indicators following CICES (Fig. 1).** Effects were separated into overall multifunctionality (displayed in the shaded areas) and multifunctionality of each of the three ecosystem-service categories, i.e., provisioning, supporting/regulating, and cultural. LRRs of the 12 single ecosystem services according to CICES shown as colored symbols. Error bars represent 95% confidence intervals based on bootstrapping. Colors of the points correspond to the respective indicators used for one distinct ecosystem service, and the shapes of the points correspond to the ecosystem-service category (Fig. 1). $N = 86$ grasslands. Source data are provided as a Source Data file.

about 20% was mineral (synthetic) fertilizer. Note that these numbers refer only to grasslands without Eco-scheme, as grasslands under Eco-scheme did not receive any organic or mineral fertilizer as required by the respective agricultural policy (Fig. 1b). Across all grasslands, mowing frequency was positively related to fertilization intensity (Spearman $rho = 0.41$, $P < 0.001$), but negatively to grazing intensity (Spearman $rho = −0.72$, $P < 0.001$; Supplementary Fig. S6). Grazing intensity and fertilization were uncorrelated ($P > 0.1$).

Supporting the GLLVM results, organic management generally showed fewer significant effects on ecosystem-service indicators than Harvest type and Eco-scheme. However, via the effect of decreasing fertilization intensity, organic management impacted eight of the 17 indicators (Fig. 4b); the respective effect sizes were, however, small in all eight cases. Eco-Scheme and Harvest type significantly affected almost all the 17 ecosystem-service indicators (16 and 12, respectively), either directly or indirectly, and with clearly bigger effect sizes than organic management.

The reductions in land-use intensity by the management aspects in turn had different subsequent effects on individual ecosystem-service indicators. For example, while the two indicators for provisioning services decreased with reduced intensity, most non-provisioning services (e.g., AM fungi, less Surface P, lower $N_2O$ emissions) increased with a reduction in management intensity (Fig. 4b). In some cases, indirect positive and negative effects on management intensity occurred simultaneously for one ecosystem-service indicator in the SEMs, leading to an insignificant overall effect of the respective management aspect in the GLLVM (Fig. 3). For instance, Eco-scheme had a negative impact on livestock presence via reduced fertilization intensity. At the same time, it increased livestock presence via reducing the cutting frequency (Fig. 4b). Thus, the overall effect of Eco-scheme on livestock presence remained insignificant (Fig. 3). The

effects of the management aspects on individual ecosystem-service indicators acting via all three measures of land-use intensity underline the general importance of land-use intensity for the majority of grassland ecosystem services studied here.

Effects of management aspects on ecosystem-service indicators not acting via the intensities of mowing, fertilization and grazing appeared as direct effects in the SEMs (Fig. 4b) and cannot be specified further. Direct effects occurred for slightly more than half of the indicators studied and were partly positive and partly negative. Most direct effects were found for Eco-scheme, followed by Harvest type. Noteworthy, Production system did not show any significant direct effect.

## Effects of management aspects on ecosystem-service multifunctionality

To evaluate overall plot-scale multifunctionality, ecosystem-service indicators were grouped according to CICES (Fig. 1), and their multifunctionality was assessed by the mean log response ratio (MLRR) of the three management aspects (see "Methods" in Supplementary Material for details). Effects on overall ecosystem-service multifunctionality were larger for Eco-scheme and Harvest Type than for the Production system, which had no significant effect (Fig. 5). Despite diverging responses of single ecosystem services to Eco-scheme and Harvest-Type pasture, overall ecosystem-service multifunctionality significantly increased with these two management practices, highlighting that grassland management can considerably affect multifunctionally at the plot scale (Fig. 5a).

For Eco-scheme and Harvest type, a clear trade-off between ecosystem-service categories was observed: Eco-scheme, as well as Harvest-type pasture, increased cultural ecosystem services on average by 43% and 73%, respectively, while decreasing provisioning

ecosystem services by −40% and −11% (Fig. 5b). Eco-scheme also promoted the category of supporting/regulating ecosystem services by on average 36%, whereas Harvest type did not significantly affect this aspect of plot-scale multifunctionality due to weak and simultaneous positive and negative effects on single ecosystem services. Note that the strong effect of pasture on the cultural ecosystem-service category was partly ruled by a strong positive effect of pasture on the ecosystem services "heritage and culture", which includes the ecosystem-service indicator "livestock presence".

## Discussion

We found the three major grassland management aspects studied to differently affect single ecosystem services and ecosystem-service multifunctionality at the plot-level. This insight in effects of grassland management, as shaped by agricultural policies and local decision-making, is required to take informed action to maintain and enhance landscape-scale multifunctionality and meeting societal needs that go beyond food production. While the Production system organic management did not affect ecosystem-service multifunctionality due to small effects on individual ecosystem services, the Eco-scheme of extensive grassland management had strong positive effects on non-provisioning ecosystem services and on multifunctionality, but decreased provisioning ecosystem services. The Harvest-type pasture was overall more beneficial for ecosystem-service multifunctionality than meadow, although different ecosystem-service indicators were promoted by either pasture or meadow. Surprisingly, we did not observe a major importance of interacting effects of the management aspects on ecosystem services. This underlines the relevance of each separate management aspect for ecosystem services and the option to freely combine these aspects to achieve the desired set of ecosystem services. Mechanistically, land-use intensity was found to be the key driver of single ecosystem services and related multifunctionality, as well as of the trade-offs observed between provisioning and especially cultural ecosystem services. Thus, the impact of the three management aspects on plot-scale ecosystem-service multifunctionality was closely related to lowering land-use intensity, albeit to very different degrees, as will be discussed in the following.

The Production system organic management appeared to play a minor role in increasing plot-level ecosystem-service multifunctionality in temperate grasslands. Organic grassland farming improved two out of the 22 indicators and did not significantly improve overall multifunctionality. However, importantly, organic farming did not have any significant negative effects on ecosystem-service indicators or multifunctionality, but showed a tendency towards lower biomass production. Interestingly, our SEM analysis did not find any direct effect of organic grassland farming on the studied ecosystem services (Fig. 4b). The overall small effect of Production system is most likely due to the rather small differences in management intensity between organic and non-organic grasslands. Yet, organic management reduced fertilization intensity in non-Eco-scheme grasslands, in which fertilization was generally allowed, by on average 32% less available N compared to non-organic management. This lower land-use intensity of organic grassland farming is connected to the ban of synthetic fertilizers and lower limits for organic fertilization. It was previously observed in other contexts[33,34] and could directly be responsible for lower $N_2O$ emissions from organic grasslands. As lower N fertilization also relates to lower P input, this might further explain the higher abundance of AM fungi in organic grassland soils, indicating lower soil P availability than in intensively fertilized conventional soils.

The small benefits of organic management observed here close the research gap concerning the portfolio of ecosystem services supplied by organic grasslands, and are in line with results from croplands, in which organic management also appeared to play a minor role for improving ecosystem services. For croplands, increases in plot-scale

ecosystem-service multifunctionality under organic management have been observed in some cases[20,35], whereas in other cases, only a small number of individual ecosystem-service indicators improved, without a clear impact on ecosystem-service multifunctionality[36,37]. The overall weak effect of organic management on plot-scale grassland multifunctionality in our study acted via the slight decrease in fertilization intensity. This highlights the importance of considering regional and/or national fertilization standards when extending our findings to other places or systems, because the use of mineral fertilizers in grassland farming and related differences in management intensity between organic vs. non-organic grasslands can vary considerably among countries[38,39]. In Switzerland, grassland management is on average mid-intensive, with less fertilizer input than for instance in Germany, France, Belgium and the Netherlands, but more than in Estonia and Czech Republic[40]. Concerning pesticide applications, its use in Switzerland is lower compared to that in Germany, France, Belgium and Czech Republic, but higher than that in the Netherlands and Estonia[40]. Thus, greater differences in land-use intensity between organic and non-organic management likely lead to greater differences in plot-level ecosystem-service provision than observed here. This was, for example, shown for intensively managed organic vs. non-organic grasslands in the Netherlands[41]. In conclusion, it can be assumed that if organic and conventional grassland systems differ significantly in management intensity, it is highly likely that this will translate into benefits for especially non-provisioning ecosystem services in the less intensively managed system.

Our study provides evidence for a strong positive effect of the Eco-scheme extensive grassland management on ecosystem-service multifunctionality, especially of cultural ecosystem services, at the plot scale. This finding is of particular relevance for managing ecosystem services in landscapes dominated by intensive grassland management. The higher plot-scale multifunctionality observed should not be mistaken as the overall best way to manage grasslands, but suggests that Eco-scheme extensive management results in overall less trade-offs among the services studied. Yet, the significant increase in ecosystem-service multifunctionality comes at the cost of the provisioning ecosystem services, introducing a strong trade-off between regulating and cultural ecosystem services on the one hand, and provisioning ecosystem services on the other hand, is in accordance with previous ecosystem services research[12–14]. While we did observe that the Eco-scheme extensive management had a positive effect on many supporting/regulating ecosystem-service indicators, four of these indicators such as weed control and $N_2$ fixation were, however, lower in Eco-scheme grasslands and promoted by more intensive management. This is in line with a previous study finding some supporting and regulating ecosystem services to be increased with land-use intensity[42]. In the latter study, these services belonged to a group of ecosystem functions and processes that depend on high nutrient input such as nitrification and earthworm abundances[43–45]. Here, further aspects of intensive management like early harvest dates (e.g., less invertebrate herbivory[46]) and weeding activities might also play a role, as farmers seem to accept more (potentially ecologically valuable) weeds in extensively managed grasslands, while focusing weed management on non-Eco-scheme (intensive) grasslands. In the case of $N_2$ fixation, the indicator used was strongly driven by biomass production, explaining the positive influence of non-Eco-scheme management and related fertilizer inputs.

In addition to lower management intensity, the studied Eco-scheme extensive management affected several ecosystem-service indicators via direct effects. While these direct effects cannot be finally explained with our assessment, we can conclude that they must act independently from current management intensity and might thus be related to land-use history, harvest and fertilizer dates, soil properties, or plant community composition. In line with the positive effect of inclination on the uptake of Eco-scheme extensive management in this

study, indirect effects acting via site history such as long-term extensive management were suggested to play a crucial role in high plant diversity in permanent grasslands in Switzerland[47]. Overall, the positive effect of Eco-scheme on plot-scale ecosystem-service multifunctionality can be seen as further evidence justifying payments to farmers for such extensively managed land, which was originally designed for biodiversity support[48].

The Harvest type, i.e., whether farmers decide to primarily graze (pasture) or mow a grassland (meadow), influenced many ecosystem-service indicators. Moreover, use as pasture promoted overall plot-scale ecosystem-service multifunctionality, especially regarding cultural ecosystem services. This effect was largely facilitated by pastures having highest values for the ecosystem-service indicator livestock presence, which has been observed to positively contribute to cultural ecosystem services[49,50], especially heritage and recreation. This case of one indicator strongly driving a measure of multifunctionality, as well as the reasoning that multifunctionality is strongly influenced by the choice of indicators, points to one of the drawbacks of the frequently applied averaging approach: The average index operates as black box regarding the contributions of single indicators[51,52]. Yet, such issues can be easily detected and put into context by assessing single ecosystem services indicators and the using the MLRR approach, which allows for a transparent assessment of the contribution of single ecosystem services to overall multifunctionality. In addition, and similar to organic versus non-organic grasslands, pastures received on average slightly less fertilizer N than meadows, which constituted a weak trade-off between provisioning and cultural ecosystem services as discussed before.

Half of the ecosystem-service indicators showing a statistically significant response benefited from use as pasture, while the other half benefited from use as meadow (Fig. 3c). These diverging effects of Harvest types are in line with previous findings. One study[53], for instance, found a positive influence of grazing on ecosystem-service multifunctionality, whereas another[54], using a different set of indicators, observed a decrease of ecosystem-service multifunctionality for grazed grasslands. Yet, other studies found grazing to have less negative or more positive effects on different aspects of grassland biodiversity compared to mowing[55,56]. This, together with effects of trampling and unselective mowing versus highly selective grazing differently shaping plant communities and their traits[57], likely explains the differences in ecosystem services observed between the two Harvest types. For example, the negative effect of grazing on earthworms was most probably due to soil compaction by trampling livestock[58], which has also been shown to potentially reduce yields[59]. On the other hand, trampling and selective grazing by livestock will have increased plant richness[47].

The fact that Harvest type significantly influenced many single ecosystem services and impacted ecosystem-service multifunctionality shows that this aspect of grassland management could indeed be an impactful lever in adjusting ecosystem-service supply in a given area, depicting a relatively easy-to-implement tool to enhance cultural ecosystem services and landscape-scale ecosystem-service multifunctionality. Our study thus implies that landscapes dominated by grassland-based livestock systems relying on outdoor grazing will deliver a set of ecosystem services less supported by livestock systems with all-year indoor feeding.

The inevitable trade-offs we observed among different sets of ecosystem services lead to the conclusion that finding a one-type-fits-all grassland management is impossible, and that multifunctionality needs to be finally achieved at the landscape scale by allocating different management to different areas within the landscape (i.e., spatial segregation of ecosystem-service production)[11]. Yet, only a well-informed use of different management approaches, as provided by our study, can optimize ecosystem-service multifunctionality as desired by the local stakeholders[15]. To do this effectively, the ecosystem-service demand and priorities of local stakeholders have to be translated into a regional management plan to optimize the ecosystem-service provision[14,60,61], as the stakeholders' rating of the importance of a given ecosystem service differs according to region and context[61]. Thus, in addition to our plot-level results, it is necessary to adopt a wider view that includes farm- and landscape-scale drivers, as, for example, not only landscape composition but also configuration is important for ecosystem-service provision.

Regarding organic management, while we observed very weak effects of this management aspect on plot-level ecosystem services, organic management system could have further effects on ecosystem services at farm and landscape scales. For example, different feed origin and composition between organic and non-organic management systems have been shown to lead to beneficial effects of organic management on ecosystem services[62]. Furthermore, Swiss organic farms were shown to have a higher proportion of land under extensive Eco-schemes than conventional farms[63], indicating farm-level benefits of organic management for biodiversity, cultural ecosystem services and ecosystem-service multifunctionality.

A further way the landscape scale should be considered when interpreting these plot-level results concerns the fact that grassland management is not distributed uniformly or randomly throughout a landscape. Eco-schemes and pastures tend to be implemented on agriculturally less favorable land, located at slopier sites[64,65]. Slopier sites have generally less favorable soil conditions and are harder to manage, resulting in less potential for intensification. The correlation of Eco-scheme and pasture with topography in the present study makes it difficult to fully disentangle management and topographic effects on ecosystem services, but on the other hand represents the real-world conditions governing the distribution of grassland types in the landscape as affected by farmers' choices and the uptake of agri-environmental policies.

Our study showed that extensive Eco-scheme management and Harvest type (pasture versus meadow) were key determinants of individual ecosystem services and ecosystem-service multifunctionality in Swiss agricultural grasslands. This highlights the impact of agricultural policies and farmers' decisions on grassland ecosystem-service supply. As no strong interacting effects of the management aspects studied were observed, these practices can be freely combined to achieve the desired set of services. This way, our plot-level outcomes can directly translate into action for landscape-scale management for ecosystem-service multifunctionality.

A main underlying driver of the improvements in ecosystem services was a decrease in overall land-use intensity, especially regarding fertilization intensity and mowing frequency. Thus, we conclude that, due to the great relevance of land-use intensity for most grassland ecosystem services, strategies and policies to support ecosystem-service multifunctionality need to regulate land-use intensity and–at the same time–need to account for resulting trade-offs such as the inevitable reduction in provisioning ecosystem services under extensive management. Overcoming these trade-offs should receive further attention in future research and practice. Because our study was focused on temperate grassland, future research should also assess management effects on the ecosystem-service multifunctionality of natural grasslands such as Savannas and prairies.

Building on plot-level assessments of management effects, such as the one carried out in the present research, investigating landscape-scale ecosystem-service multifunctionality considering both landscape composition and configuration, while suggesting new or alternative ways to manage grasslands (e.g., increasing the plant diversity of swards[12]), could have the potential to additionally benefit the portfolio of ecosystem services provided by agricultural landscapes. Meanwhile, our plot-level results clearly suggest that diversifying grassland management where this is currently rather homogeneous across farms and landscapes would be an important and effective first step to increase

ecosystem-service multifunctionality for sustainable grassland systems.

## Methods

This research complies with all relevant ethical regulations and ETH Zürich's legal service approved data collection, use and storage. The latter was also part of the declaration of consent that was signed by the farmers providing information on their grassland management.

### Study area, local management practices, and sites

Measurements were carried out on permanent grassland, i.e., grassland not included in any crop rotation, in the Swiss Canton of Solothurn. This region presents a wide range of environmental conditions and stretches from the intensively managed Swiss lowlands (400–500 m a.s.l) in the South to the undulating Jura mountains (up to 1445 m a.s.l.). Agriculture in the canton is characterized by a high share of permanent grasslands (50% of the agriculturally used area[66]), with comparably small parcels (average 0.9 ha[53]) and farms (on average 23 ha) slightly higher than the national average (20.86 ha[67]).

On organically managed grasslands, the use of mineral fertilizers and synthetic pesticides is forbidden. In Switzerland, the maximum allowed amount of organic fertilizers applied per year and hectare is somewhat lower than for non-organic grassland farming (135 vs. 162 kg available N for all intensive (non-Eco-scheme) grasslands of a farm at low elevations[68]). As organic management is a farm-wide scheme and further guidelines exist, amongst others, regarding mowing and hay processing techniques, animal feed, fertilizer trade among farms, and access of livestock to outdoor areas[68]. In the Canton of Solothurn, 18% of grassland area is managed organically[66]. The studied Eco-schemes depicts an agri-environmental scheme requiring extensive grassland management with at least one harvest every year either by cutting (extensive meadows) or grazing (extensive pastures). Extensive management refers mainly to the ban of fertilization but can also include further regulations such as a delayed first date of cutting of meadows. In Switzerland, as part of the cross-compliance requirements for the eligibility to direct payments, a minimum of 7% of the utilized agricultural area of the farm must be dedicated to Eco-schemes but farmers can voluntarily register additional land beyond this minimum share. Extensively managed grasslands do not receive any fertilizer. Extensive meadows are further allowed to be mown only starting from a defined date (i.e., mid-June in lowland regions). In the Canton of Solothurn, our study area, extensively managed grasslands comprise 33% of the total permanent grassland area[66]. For the Harvest type, we chose the two dominant grassland types occurring in Central Europe. While meadows are predominately mown, with some occasional grazing such as at the end of the growing season, pastures are mainly grazed and rarely cut. This differentiation follows the official typology for Swiss grasslands and was confirmed by farmer interviews (Supplementary Table S1).

Grassland plots were selected as described in the Supplementary material (Supplementary Methods). The result was a set of 86 grassland parcels, belonging to 36 farms (18 organic and 18 non-organic) across the Canton, spanning an elevational gradient from 435 to 1145 m a.s.l. (see also Supplementary Fig. S1 for an overview of the number of farms and plots per eco-region of the Canton, and Supplementary Table S1 for the number of plots included for each single combination of management aspects).

Several environmental factors were measured to account for potentially confounding effects of the local environment. Sand content and pH was measured from the soil samples taken in June 2020. Soil fractions were measured with a SP 2000 Robotic Clay Fraction Analyzer (Skalar Analytical B.V), and soil pH was measured potentiometrically from a water suspension of soil. The elevation of the plots was derived from a Digital Elevation Model (DEM) of the Copernicus Land Monitoring Service of the European Environment Agency[69] at a

resolution of 25 m. From the DEM, northness, representing the orientation of the raster cell to the north, with +1 indicating north, and −1 south, was calculated. In QGIS.org, aspect of the land is in radians, and subsequently the cosine of this grid was computed to provide the northness. The inclination of each plot was assessed using the cell phone appliance Clinometer plaincode™.

### Management interviews with farmers

A questionnaire survey was conducted with farmers in January/February 2021 and 2022 in order to collect detailed information on the management of each investigated grassland plot. The information included grazing dates, number, age, and type of animals, as well as timing, amounts and nature of fertilizer applications. The grazing information was used to calculate the average livestock unit days ha$^{-1}$ year$^{-1}$ for each plot over the 2 years. From the information about amount and type of fertilizer, the total plant-available fertilizer N ha$^{-1}$ year$^{-1}$ was calculated based on information from ref. 70 about available N contents of the different organic fertilizers. Mineral fertilizer N was set to 100% available. The interviews also included questions about weed control measures (pesticide or mechanical; Supplementary Table S1).

### Measuring ecosystem-service indicators

In 2020 and 2021, intensive field and lab work were carried out to measure the 22 ecosystem-service indicators (Fig. 1), presenting twelve ecosystem services according to the CICES typology[29]. Regarding the measurement of ecosystem-service indicators, only the most relevant information is given here. Further details on these measurements and related analyses can be found in Supplementary Methods. The respective units of the measured ecosystem-service indicators are given in Supplementary Table S3.

In June 2020, a first soil sampling campaign was conducted to measure heavy metals, organic carbon stocks, microbial biomass carbon, and to determine the proportions of fungal guilds. Per plot, 20 soil samples along two 20 m transects were taken to a depth of 20 cm and pooled for subsequent analysis. Copper and zinc concentrations were analyzed from 2-mm sieved and air-dried soil using ICP-OS (5110 VDV ICP-OS, Agilent, Santa Clara, CA, US), divided by the respective reference values for Swiss soils, and the highest value of the two metal concentrations per plot was used for the ecosystem-service indicator heavy metals. Soil organic carbon was measured via sulfochromic oxidation[71], and carbon stocks were calculated by multiplying organic carbon concentration with bulk density from 5 to 10 cm, which was measured as described below. Microbial biomass carbon was determined via chloroform fumigation[72,73]. For determination of the proportion of fungal guilds, specifically proportion of arbuscular mycorrhizal fungi (AMF) DNA, plant pathogenic fungi, and iconic fungi, DNA extracted from the soil samples was used for sequencing the fungal ITS region on an Illumina platform (Illumina, San Diego, CA, USA). DNA extraction, sequencing and bioinformatic processing was performed following ref. 74, but see Supplementary Methods for details. Information about fungal guild membership of fungal taxa was identified using FunGUILD[75] for AM fungi and plant pathogenic fungi. Taxa belonging to CHEGD taxa, which include the often particularly colorful grassland macrofungi of high conservational value[76], were identified as indicator for iconic fungi.

In August and September 2020, a second soil sampling campaign was carried out to measure root biomass, bulk density for soil compaction, and soil surface phosphorus concentrations. Root biomass was assessed by washing and sieving soil cores from 0 to 5 cm depth, from three pooled samples per parcel. To determine bulk density, the fine soil stock (FSS, g cm$^3$) was calculated according to ref. 77 and used together with clay content to calculate packing density[78] as a measure for soil compaction, which is closely related to infiltration capacity, using three pooled soil samples from 5 to 10 cm depth per plot.

For surface soil phosphorus (P) concentrations, we used ten pooled shallow soil samples (1.5 cm deep) per plot, representing the stratum of soil P which is particularly at risk of erosion and thus depicts a potential eutrophication risk for freshwater ecosystems. Water-extractable soil phosphorus was measured photometrically (Evolution 220 with Cetac ASX-520, Thermo Fisher Scientific, Waltham, MA, USA).

Between the beginning of May and mid-June 2021, vegetation and earthworm surveys were conducted. For vascular plant species richness, all plant species occurring at two 2 m × 2 m quadrats (20 m apart from each other, each 10 m from the plot center) were recorded and summed for a total richness in plant species. The number of edible plant species was calculated based on the vegetation survey (the two 2 m × 2 m quadrats) and literature information[79–81]. Potential nectar provision was estimated using the cover of plant species from the vegetation surveys and data on nectar provision per species from the literature[82,83]. The number of agricultural weed plants (or of dense patches for clonal plants) was recorded along two transects per site. Leaf damage by herbivorous arthropods was assessed by sampling leaves in the field; one legume, grass, and herb leaf each (if available), every 50 cm along two perpendicular 20 m transects. subsequent visual examination of damage, and percentage of damaged leaves was used as the ecosystem-service indicator. Aboveground plant biomass was sampled repeatedly on the plots. In pastures, grazing exclusion cages were installed prior to grazing, and biomass could be sampled at 2 cm above the surface. The biomass sample taken closest to the first date of use as indicated by the farmers in a management survey was analyzed for digestibility (i.e., digestible organic matter) via enzymatic digestion in rumen fluid according to ref. 84. For aboveground biomass yield, vegetation biomass was sampled close to a reference date set to end of May (day of year, DOY, 146). As some plots were sampled later or earlier, either due to displacement of the grazing exclusion cages by cow activity or to logistic reasons, biomass was corrected for sampling date. To this end, biomass weight was divided by the temperature-degree sum until sampling date following the approach described in ref. 85. Symbiotic $N_2$ fixation in biomass harvested close to DOY 146 was calculated based on the aboveground biomass as described above and by considering identity and cover of occurring legume species. N content of the legume species occurring on each plot was measured and used together with the modeled mass percent of the respective legume and the biomass yield measure to calculate an index for N fixation (Supplementary Methods). To assess earthworm abundances, three soil pits (30 × 30 × 30 cm) were dug out and the excavated material was checked manually, and number of earthworm individuals were counted.

Esthetic appreciation of the plant community was derived from standardized pictures of each plot taken prior to the vegetation surveys. An online survey asking people for their personal perception of the esthetic quality of the respective grassland plant community on a 5-point Likert scale from attractive to unattractive was set up with QuestionPro (QuestionPro Inc, Austin, TX, USA), and widely distributed over e-mail and social media, with finally 414 respondents. The mean esthetic rating per plot was used as a value of esthetic appreciation. $N_2O$ emissions were calculated according to the IPCC guidelines[86], using fertilizer data from the management interviews and Switzerland-specific information on livestock[70] to estimate the amount of N excreted by grazing animals. Information on N inputs was also used to estimate potential nitrate leaching using a tool developed by the UK Environment Agency accounting for fertilizer N and animal excreta as sources for nitrate leaching[87].

## Data analyses

Data were analyzed on three levels. First, ecosystem-service indicators were analyzed using multivariate regression, which allowed to jointly model all 22 indicators without loss of information while considering the correlations among them (see ref. 88 for a discussion of advantages). To allow for direct comparisons among model terms, indicators were normalized to a common scale by dividing each of the 22 response variables by their maximum[87,89]. To attain a multivariate normal distribution of residuals, some of the indicators had to be log-transformed and indicators, for which small values were regarded to have positive benefits (e.g., nitrate leaching), were reversed by subtracting these indicators' maximum value from their respective values (see Supplementary Methods for details on these transformations). Using general linear latent variable models (GLLVM[32]), the response matrix of ecosystem-service indicators was then regressed on the management aspects, namely Production system (factor with two levels: organic, non-organic), Eco-scheme (yes, no), and Harvest type (pasture, meadow). All regressions implied a Gaussian link and two latent variables, as models with more than two latent variables did often not converge. Where models converged, more than two latent variables did not change the fixed estimates and their standard errors. "The latent variables can be thought of as ordination scores, capturing the main axes of covariation of responses after controlling for observed predictors" (adapted from ref. 32). We included a random factor "farm pair", with a level each to include all plots within a pair, to account for the blocking structure regarding pairs of organic/non-organic farms in the sampling design. We also tested a random factor "farm", with a level each to include all plots within a farm; however, this random variance was always estimated to be zero and the random "farm" term was omitted from all models. The most parsimonious model was identified in the following way: we first ran a simple model including only the main effects of the three management aspects. Then we used forward selection under the second-order Akaike Information Criterion (AICc)[90] to determine, which of the five environmental variables (pH, sand content, elevation, inclination, and northness; each transformed to standard deviation scale) resulted in a most parsimonious model. Then, we ran this resulting model including any of the two-way interactions between the three management aspects (see Supplementary Table S2). It turned out that a model with main effects and three of the environmental variables was most parsimonious (AICc = −2271.3). Given this outcome, we present the main effects in the results section and the effects of the environmental variables in Supplementary Fig. S2.

Second, to gain insights on the drivers of the observed effects of the three management aspects on the ecosystem-service indicators, structural equation models (SEMs) were calculated in the package lavaan[91]. In a first step, the effects of the topographic variables, i.e., elevation, northness and inclination, were tested on Eco-scheme and Harvest type. This was done because topography might have influenced farmers' decision on how to manage the grassland. This was, however, not necessary for Production system, as meaningful differences in this factor had been eliminated due to plot selection (the blocking factor) in the sampling design. In a second step, the effect of the management aspects was tested on the three key measures of grassland management related to land-use intensity[31]: fertilizer N (available N: sum of organic and synthetic fertilizer, kg ha$^{-1}$ year$^{-1}$), number of cuts (year$^{-1}$), and grazing intensity (livestock units days ha$^{-1}$ year$^{-1}$), derived from the management interviews. In a third step, the direct effect paths from the three management aspects to an ecosystem-service indicator as well as indirect effect paths via the three management intensity variables were tested. The model structure is shown in Fig. 4a; the model was calculated separately for each indicator showing any statistically significant effect of management aspects in the GLLVM model.

Third, to assess the effect of the different management aspects on overall plot-scale multifunctionality, we used the approach suggested by Suter et al.[92], which is based on the mean log response ratio (LRR) across ecosystem-service indicators for a given treatment comparison (e.g., organic vs. non-organic). In the context of multifunctionality,

LRRs are a very useful measure because they are dimensionless effect sizes comparable among ecosystem services, can be calculated for different conditions (as in meta-analyses), and have particularly desirable statistical properties[93]. Moreover, the mean of several ecosystem-service indicators' LRR (MLLR) between any two types of management (e.g., Eco-scheme yes versus no) has an intuitive meaning in that a greater number of indicators with higher LRRs reflect enhanced overall performance and therefore greater multifunctionality of one management type compared to another. Finally, the Euler's number $e$ raised to the power of the MLRR gives an overall effect size on the linear scale.

We preferred the MLRR as a measure of multifunctionality[92] over the most widely used averaging approach, where ecosystem-service indicators are averaged to result in one value of ecosystem-service multifunctionality per plot. This preference was motivated by at least two reasons. The first relates to interpretability. While the MLRR can be interpreted as an overall effect size derived from distinctly interpretable LRRs (similar to meta-analysis), a simple average of ecosystem-service indicators has little general meaning, even when individual variables are adjusted to a common scale. This is because variables with very different units and distributions are pooled (for example: masses, contents, counts, some of which typically have non-normal distributions), making it difficult to interpret such an index. In particular, effect sizes derived from such indices cannot be interpreted reasonably. The second reason for choosing the MLRR is that it allows for a transparent assessment of the contribution of each ecosystem service to the overall mean (compare Fig. 5). In contrast, the averaging approach comes with a loss of information, as ecosystem-service indicators can have (equal) opposite values or equivalent values, yet both cases will result in the same average metric. The impact of individual ecosystem-service indicators on such an average is generally not assessable. Finally, with our approach of using the MLLR, we follow recent demands to focus on effect sizes and their confidence intervals, rather than means and null-hypothesis significance testing[94] See also ref. 88 for a discussion of different multifunctionality metrics.

Here, given the outcome of the GLLVM, LRRs were calculated comparing organic versus non-organic, Eco-scheme yes versus no, and pasture versus meadow. As the main effects of the three management aspects did not or only marginally change, when the environmental variables were added to the model (compare Fig. 3 and Supplementary Fig. S4), we avoided integrating these to the calculation of the MLRR. In particular, we refrain of using residuals from an initial regression of ecosystem-service indicators on the environmental variables to control for potential confounding because this procedure leads to biased model estimates[95]. For the calculation of the MLRR, LRRs of ecosystem-service indicators for which small values were regarded to have positive benefits were multiplied by −1. Then, the LRRs of indicators contributing to the same CICES-ecosystem service were averaged across the 12 CICES-ecosystem services (Fig. 1) and the MLRR was then calculated as the mean across the LRRs per CICES-ecosystem service. In doing so, ecosystem-service indicators informing about different aspects of one ecosystem service were downweighted to avoid over-representation of one CICES-ecosystem service over the others. Note that the aggregation of ecosystem-service indicators to a CICES-ecosystem service is equivalent to taking the average across all LRRs and weighting indicators by 1 divided by the number of respective indicators per CICES-ecosystem service. We chose to use the CICES framework as it is widely used in science and practice and thus allows for a better comparability of the results to other work. The MLRR was calculated across all CICES-ecosystem services and for each of the three ecosystem-service categories "provisioning", "supporting/regulating", and "cultural" to highlight potential trade-offs in the overall performance of management aspects. Finally, the 95% confidence interval to the MLRR was calculated by bootstrapping[96] (see Supplementary Methods and the Supplementary Code for details to averaging of LRRs

across CICES-ecosystem services and the bootstrapping). All analyses were done using the statistical software R, version 4.2.0[97] and the package gllvm[32]. Further details concerning the statistical analyses can be found in Supplementary Methods. To support our results on overall multifunctionality based on the MLRR, we evaluated the effects of the management aspects (organic vs. non-organic, Eco-scheme extensive yes vs. no, and pasture vs. meadow) on multifunctionality using two further methods. First, we calculated a mean multifunctionality index using the estimates of the GLLVM as shown in Fig. 3, and second, we calculated multifunctionality using the averaging approach. Using these two alternative approaches, we found the multifunctionality results to be highly similar to the MLRR. Based on these similar outcomes from two alternative methods, we conclude that our results for the MLRR are not only well interpretable but also highly reliable.

### Reporting summary
Further information on research design is available in the Nature Portfolio Reporting Summary linked to this article.

## Data availability
The ecosystem-service indicator, environmental, and management data generated in this study have been deposited in the ETH research collection under accession code ethz-b-000663689. The sequencing data is deposited in the European Nucleotide Archive under the accession number PRJEB72428. Further databases used in this study: Unite v.83 database (Kõljalg et al.[98]), FUNGuild database (Nguyen et al.[75], Data on Nectar availability from Baude et al.[82] and Filipiak et al.[83]. The digital elevation model used is from the European Union (2018) Copernicus Land Monitoring Service, European Environment Agency (EEA) [https://doi.org/10.5270/ESA-c5d3d65] (accessed 12.25.20). Source data are provided with this paper.

## Code availability
R Codes used for data preparation, to run the GLLVM, the SEMs, and the bootstrapping procedure for the confidence intervals to the MLRR is provided in the Supplementary Material. For all other analyses, available R packages were used.

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

## Acknowledgements

The authors thank the farmers of the grasslands, all persons involved in field and lab work, especially Eliana Mohn, Friederike Meyer and Jannis Weil, and all people involved in project administration (Anna Gilgen) and scientific discussions, especially Olivier Huguenin-Elie, Peter Manning,

Sara D. Leonhardt, Solen Le Clec'h and Eric Allan. We acknowledge funding from the Mercator Foundation Switzerland (15398; V.H.K., A.L., and N.E.-B.), the Foundation Sur-la-Croix (V.H.K. and A.L., no grant number), and the Pancivis Foundation (PAN 2019/31; V.H.K. and M.H.). Finally, we thank the anonymous reviewers whose feedback improved this work.

## Author contributions

V.H.K. and A.L. conceived the idea and developed the project with inputs from F.J.R. and N.B.; F.J.R. was responsible for data generation with input and support by M.H., R.F.-C. and V.H.K.; F.J.R. and M.S. analyzed the data with input from V.H.K.; F.J.R. wrote the manuscript, led by V.H.K. and M.S., to which N.B., N.E.-B., P.J., A.L., M.H. and R.F.-C. contributed advice and comments, and gave final approval for publication.

## Funding

## Competing interests

The authors declare no competing interests.
