## [Peer Review File · Nature Communications]

Reviewers' Comments:

Reviewer #1:

Remarks to the Author:

Review comments for "Farming for ecosystem services: effects of major management aspects on grassland multifunctionality"

Summary: This manuscript takes a holistic approach to assess the impacts of three major agricultural grassland management practices, including organic farming, extensive management and harvest type, on a list of 22 indicators for ecosystem services (ES). The authors revealed important tradeoffs among ES indicators, in which organic farming only showed marginal impacts on ES provision with two services improved, whereas the extensive management led to increases in the ES multifunctionality (mostly cultural and regulating ES but at the expense of provisioning ES). Grazing (using grasslands as pastures) showed positive impacts on the provision of cultural ES. The manuscript concludes the importance of adopting and combining multiple grassland practices at the landscape scale, given their tradeoffs and differential performance.

Overall, the manuscript is very well written with professionally-crafted and clearly presented figures. I am very impressed by the large amount of datasets that the authors have assembled to comprehensively demonstrate the multi-dimensional effects of major grassland management that are often implemented together in practice. I think this work has potential to enrich the existing literature on landscape management and ES multifunctionality, especially for the region being studied and also inclusion of diverse ES indicators. However, I do have several major concerns pertinent to the conceptual framing, methodologies, and result presentation and interpretations, which hopefully could be helpful for the authors to consider and revise this work. Below I start with substantive comments, followed by more detailed minor suggestions.

Substantive comments:

(1) Framing: The paper argues that the novelty of this work lies in understanding effects of grassland management on a broad range of ES and associated multifunctionality, which have not been comprehensively investigated in the literature (e.g., L38). However, this argument is largely overstated. As I pointed below, there is a plethora of research across different geographic regions and systems that have studied this issue. Nonetheless, what I think is unique about this paper is the assessment of 'sequentially integrated and interactive' grassland management (e.g., grazing/mowing after organic/extensive management, as explained in study design and illustrated in Fig. 1), as well as interactions among the implementation of these different management practices and their consequences for multifunctionality. The authors may consider some reframing in the Introduction/Discussion to reposition and highlight such novel contribution.

(2) Methodology: One possible concern I had regarding the statistical analyses (i.e., GLLVM) is that, given the sampling design and grassland site selection, I think a hierarchical linear regression model would be more appropriate. Essentially, the effects of eco-scheme and harvest type (along with their two-way interactions) are nested within the production system. In addition, given the paired sites of organic vs. non-organic, a blocking factor needs to be included in the model for proper testing and comparisons. In addition, I would also suspect that other factors such as climatic, vegetation, and soil variables (e.g., soil type, texture, nutrient status, grassland diversity and composition) could also affect ecosystem functions and services. In your models of GLLVM and SEM, have you thought about incorporating and teasing apart any potential confounding factors from climate, soil and vegetation variables, or extent to which the effects of management are manifested through these additional important factors?

(3) Results presentation: Given the design of the study, the most intriguing part of the results that remains less well emphasized and represented is the interactions among these different management types (examples as shown in Fig. S1.3), in particular for the ES multifunctionality. Because these management practices are often implemented together (as also stated by the authors), the more meaningful and potentially management-relevant findings would be how the combination of management would affect/alter each category of ES, as well as ES multifunctionality.

(4) Discussion and Interpretation: Much of the current form of discussion relies on re-iteration the patterns from the Result section, without providing much interpretation on what factors or processes would mechanistically drive revealed management effects. Please see examples of my detailed comments below where additional, more in-depth interpretations would be helpful, especially regarding effects of eco-scheme and harvest type).

Detailed comments:

L6: consider replacing "either using..." with "harvest type" which has been used in the rest of the manuscript.

L33: Why temperate grasslands and their ES reply on grazing or mowing? How about grasslands in other biomes? Grasslands can also provide many ES without moving or grazing. It needs to be clarified.

L38: This statement is largely overstated. There has been a large number of papers focusing on grassland management on ecosystem multifunctionality, either individual land management or multiple practices and across different biomes, for example, Neyret et al. 2021; 2023; Allen et al. 2015; Wang et al. 2019; Schils et al.2022, just to name a few.

L45: typo? "1farmer's choice"

L60: "yes versus no" seems not necessary.

L65-66: similar to the comment above, this is likely an overstated argument. There has been quite a lot of studies that have investigated consequences of land-use intensity in grassland multifunctionality.

L172-173: Any idea of the relative importance (or proportion of variance explained) between direct vs. indirect effects?

L273: Given the interpretation of the results, this statement may need to be limited to the areas (e.g., Swiss grasslands) where the background fertilization/pesticide use is much lower for non-organic farming grasslands. Otherwise, broad argument as such could be potentially misleading.

L302: Please provide explanations on why certain ES (such as weed control and N fixation) are promoted by land-use intensity?

L325-326: Sorry if I am confused here. I thought that based on the methods, the mean LRR was calculated using the mean values of LRR for each indicator, which essentially occurs to me as a simple average approach? I'd appreciate if you can further clarify how the MLRR approach differs from and helps address the black box regarding the contributions of single indicators.

L330-332: Please provide any possible mechanisms on the diverging effects of harvest types.

L382: A study map of the grassland ecosystems, as well as locations/(possibly photos) of the study plots would be very informative – e.g., to demonstrate spatial gradient of these plots.

L452-474: Some sampling details seem to be missing. For example, how many 2x2 m quadrats per plot were used for sampling vascular plant species richness, and are they randomly placed in each plot or along the transect? Were the edible plant and nectar plant also sampled from the 2x2 m quadrats, and if not, how were they sampled? For leave damages, how were the leaves sampled and collected, and how much were sampled for each plots?

L483: Please specify what tools used for quantifying potential N leaching.

L497: Please specify what are the two specific latent variables in the GLLVM and the justifications.

L535: How the bootstrap was performed to calculate MLRR can be further clarified; for example, in

Supplementary S2.1, it mentioned to bootstrap 1000 times from the replicated plots (i.e., 86 plots total). Can you specify what percentage of samples was randomly selected from each iteration of bootstrapping?

Fig. 1b: For the graphics of production system – organic, do you mean a “x” (instead of check mark) on the synthetic pesticides icon?

References cited in the comment

Neyret, M., Fischer, M., Allan, E., Hölzel, N., Klaus, V.H., Kleinebecker, T., Krauss, J., Le Provost, G., Peter, S., Schenk, N. and Simons, N.K., 2021. Assessing the impact of grassland management on landscape multifunctionality. *Ecosystem services*, 52, p.101366.

Neyret, M., Peter, S., Le Provost, G., Boch, S., Boesing, A.L., Bullock, J.M., Hölzel, N., Klaus, V.H., Kleinebecker, T., Krauss, J. and Müller, J., 2023. Landscape management strategies for multifunctionality and social equity. *Nature Sustainability*, pp.1-13.

Allan, E., Manning, P., Alt, F., Binkenstein, J., Blaser, S., Blüthgen, N., Böhm, S., Grassein, F., Hölzel, N., Klaus, V.H. and Kleinebecker, T., 2015. Land use intensification alters ecosystem multifunctionality via loss of biodiversity and changes to functional composition. *Ecology letters*, 18(8), pp.834-843.

Wang, L., Delgado-Baquerizo, M., Wang, D., Isbell, F., Liu, J., Feng, C., Liu, J., Zhong, Z., Zhu, H., Yuan, X. and Chang, Q., 2019. Diversifying livestock promotes multidiversity and multifunctionality in managed grasslands. *Proceedings of the National Academy of Sciences*, 116(13), pp.6187-6192.

Schils, R.L., Bufe, C., Rhymer, C.M., Francksen, R.M., Klaus, V.H., Abdalla, M., Milazzo, F., Lellei-Kovács, E., ten Berge, H., Bertora, C. and Chodkiewicz, A., 2022. Permanent grasslands in Europe: Land use change and intensification decrease their multifunctionality. *Agriculture, Ecosystems & Environment*, 330, p.107891.

Reviewer #2:

Remarks to the Author:

The authors evaluated the impacts of three grassland management practices (i.e., organic farming, eco-scheme extensive management, and grazing vs mowing) on the ecosystem services (ES) and their multifunctionality at the plot level in the Canton of Solothurn in the northwest of Switzerland. The study selected 22 ES indicators covering provisioning, supporting/regulating, and cultural services, and used multivariate regression and structure equation modeling for statistical analyses. The main findings were: (1) organic farming did not improve ES multifunctionality as expected; (2) eco-scheme extensive management improved the overall ES multifunctionality most substantially – increasing a number of regulating and cultural services at the expense of provisioning services; (3) the different management practices affected ES mainly by changing land use intensity.

The topic of the study, ecosystem service multifunctionality, is certainly important to sustainable ecosystem management, sustainable land planning, and landscape/regional sustainability. The dataset looks unique and interesting. The statistical methods seem appropriate for addressing the research questions (although they were not explicitly stated in the manuscript). The findings about the effects of different grassland management practices on individual ES and their tradeoffs generally corroborated those from many previous studies in different parts of the world. So, they are not new, but just similar findings from a Swiss region. Nevertheless, the three main findings are interesting and may potentially have important implications for grassland management in the study region. However, I am afraid that some of the major conclusions and recommendations were not justified or supported by their results. The main reasons for this are some fundamental problems with the conceptualization and analysis framework of the study. Let me summarize them briefly below:

1. All analysis was conducted at the plot level (i.e., "grassland parcels"), but the focus of the study was about ES-multifunctionality. This seems a major reason why the results were not really addressing the research objectives. At the plot or site level, ES tradeoffs are inevitable and easily understandable because each plot (or site) may be used mainly for a particular purpose (or one primary ES or a group of ES). Thus, ecosystem service multifunctionality needs to be studied and achieved at the landscape (as well as regional) scales. This seems the dominant view in the current literature in landscape multifunctionality and landscape sustainability. Indeed, it is a key tenet of the emerging research field, landscape sustainability science (<https://link.springer.com/article/10.1007/s10980-013-9894-9>). Unfortunately, although the manuscript mentioned "landscape-scale sustainability" once in passing, it's not clear at all how the plot-level ES multifunctionality is related or relevant to landscape sustainability.

2. Related to the above, most of the results and conclusions are confusing and, sometimes, nonsensical, because there was not an explicit spatial scale attached to it (e.g., plot or landscape). Diversifying ES at a "plot" level sounds unpractical as a plot is rather small in size. It seems more appropriate and much more informative to conduct the management strategies-ES multifunctionality analysis on multiple scales: from plots/sites to landscapes/regions. This way, the plot-level and landscape level ES multifunctionality can be distinguished and compared. More importantly, the impacts of landscape composition and configuration on ES multifunctionality – a key research topic in landscape sustainability research – can be investigated.

3. Simply counting the number of ES as the measure of multifunctionality may lead to ecologically unwarranted conclusions. For example, ES related to water/soil pollution ought to be emphasized because they are essential to ecosystem functions and human health. Also, eco-scheme extensive management did not use fertilizers and pesticides and have low levels of land use intensity, and consequently a number of regulating and cultural ES increased, at the expense of provisioning ES. But, can a region be sustainable without local food production?

4. The three management strategies did not seem really "independent" as they claimed because, for example, Eco-scheme had mowing whereas pastures also had mowing (vice versa – meadows also had grazing). Thus, the statistical analysis did not consider these complications. Thus, I suggest that you will deal explicitly with the problem of data interdependence (e.g., considering only pastures without mowing and meadows without grazing).

5. In the Discussion section, the authors stated: "the ES demand and priorities of local stakeholders have to be combined with plot-level assessments of ES from studies like here in a regional management plan to optimize the ES-provision on the landscape scale ... Thus, ... it is necessary to adopt a wider view that includes farm and landscape scale." I agree, but optimizing ES multifunctionality at the landscape scale requires the consideration of both composition and configuration (spatial arrangement). The latter was not studied in this study.

Then, they concluded: "diversifying grassland management where this is currently rather homogeneous across farms and landscapes would be an important and effective step to increase ES-multifunctionality for sustainable grassland farming." Well, again, this sounds right, but ES diversification and landscape sustainability require the consideration of not only the spatial pattern of ES, but also their tradeoffs across spatial and temporal scales.

6. The effects of the selection of ES indicators on the results have been reported in recent studies, which should be discussed in the context of this study.

7. The use of the term, "farming", in this paper may be reconsidered because its narrow definition usually refers to cultivation or crop production.

Reviewer #3:

Remarks to the Author:

Dear Editor,

This manuscript provides a comparison between different management systems for the Swiss Alps,

which is potentially representative of European temperate grasslands. The study is based in 86 plots quite evenly distributed between organic and non-organic, extensive and intensive, pastures and meadows. Authors analyse the contribution of each of the 8 possible combinations of management types on 22 ES indicators. The text is well written and the conclusions are supported by the results, which show quite strong evidence for their findings (main differences between production systems are intensive versus extensive and secondarily pasture vs meadow rather than organic vs non-organic).

However, given the number of studies currently published on temperate grasslands multifunctionality, the novelty of the approach and findings needs to be better justified.

A few points for improvement of the manuscript remain:

1. The text of the abstract could be improved to better capture readers' attention. For example, it needs a better introduction and justification of the novelty of the study and their findings.
2. The term eco-scheme could simply be named extensive/intensive, which would avoid misunderstandings, given that the term eco-scheme is very broad and can imply other aspects too.
3. The extensive eco-scheme does not seem to be independent from organic, given that all organic farms are extensive and extensively managed grasslands do not receive any fertilizer in this study. It is not clear how authors account for this interaction.
4. Somehow the text seems to avoid supporting organic farming practices, as not having additional positive effect. However, these results can also be interpreted as organic farming not having a negative effect on provisioning ES, which is often the key criticism of this management system. I would suggest to hence present these results in a more neutral tone.
5. The conclusion section would benefit for more concrete statements. For example, instead of "regulate land use intensity", authors could point out to which levels of land use intensity should be recommended, and how to deal with trade-offs.

Minor details:

- L. 34 it needs to be explained what entails intensification of agricultural management in the context of temperate grasslands, and why is that a threat to ES-multifunctionality
fig 5. Legend typo in "active recuperation"

Methods:

- the use of MLRR to calculate multifunctionality needs to be better justified, at least provide the reference given in the Sup. Material.
- Also, could you provide a reference for the standardization method used? In principle, the standardization based on normalization that scales the data between 0 and 1 involves: $X \text{ scaled} = (X - X_{\min}) / (X_{\max} - X_{\min})$

Reviewer #1 (Remarks to the Author):

Review comments for "Farming for ecosystem services: effects of major management aspects on grassland multifunctionality"

Summary:

This manuscript takes a holistic approach to assess the impacts of three major agricultural grassland management practices, including organic farming, extensive management and harvest type, on a list of 22 indicators for ecosystem services (ES). The authors revealed important tradeoffs among ES indicators, in which organic farming only showed marginal impacts on ES provision with two services improved, whereas the extensive management led to increases in the ES multifunctionality (mostly cultural and regulating ES but at the expense of provisioning ES). Grazing (using grasslands as pastures) showed positive impacts on the provision of cultural ES. The manuscript concludes the importance of adopting and combining multiple grassland practices at the landscape scale, given their tradeoffs and differential performance. Overall, the manuscript is very well written with professionally-crafted and clearly presented figures. I am very impressed by the large amount of datasets that the authors have assembled to comprehensively demonstrate the multi-dimensional effects of major grassland management that are often implemented together in practice. I think this work has potential to enrich the existing literature on landscape management and ES multifunctionality, especially for the region being studied and also inclusion of diverse ES indicators. However, I do have several major concerns pertinent to the conceptual framing, methodologies, and result presentation and interpretations, which hopefully could be helpful for the authors to consider and revise this work. Below I start with substantive comments, followed by more detailed minor suggestions.

Response: Thank you for your positive assessment and the constructive comments and suggestions. We have addressed the issues raised and have adapted our methods to accommodate the suggestions.

Our point-by-point answers can be found below. In quotes from the manuscript text, new or altered passages are highlighted in yellow.

Reviewer's comments:	Authors comments
(1) Framing: The paper argues that the novelty of this work lies in understanding effects of grassland management on a broad range of ES and associated multifunctionality, which have not been comprehensively investigated in the literature (e.g., L38). However, this argument is largely overstated. As I	Thank you for this comment, and for outlining the strength of our paper. We agree that the key aspect of the novelty of our work is in analysing not only the single but also the interactive effects of the three independent grassland management aspects. We further agree to refer more obviously to previous studies on grassland ecosystems services. As a consequence, we have widely rephrased the introduction and relativised the claim about the scarcity of studies investigating a

pointed below, there is a plethora of research across different geographic regions and systems that have studied this issue. Nonetheless, what I think is unique about this paper is the assessment of 'sequentially integrated and interactive' grassland management (e.g., grazing/mowing after organic/extensive management, as explained in study design and illustrated in Fig. 1), as well as interactions among the implementation of these different management practices and their consequences for multifunctionality. The authors may consider some reframing in the Introduction/Discussion to reposition and highlight such novel contribution.

wide range of grassland ES (see also our answer below to the comment concerning lines 38 and 65-66). We also updated the references, to which we refer in this manuscript in this regard, and we adjusted the framing of our work, in the Abstract.

It now reads:

“We assessed how **employing and combining three** widespread aspects of agricultural grassland management, namely i) organic production system, ii) eco-scheme ‘extensive management’, and iii) **harvest type (pasture vs. meadow)**, can enhance plot-level ES and respective multifunctionality, based on 22 ES-indicators.” (L 5)

And in the Introduction, in several places:

“Before introducing agricultural policies that promote certain farming practices, the effectiveness of these practices needs to be assessed in terms of their **individual and combined** environmental benefits.” (L 21)

“Here, we tested three aspects of grassland management **that are widespread in their adoption and implemented independently from but alongside each other**, for their ability to increase ES-multifunctionality.” (L 49)

“To address the question of how these three widespread management aspects, as well as the interactions among them, influence grassland ES multifunctionality, we assessed 22 ES-indicators in 86 managed grasslands in the Canton of Solothurn, Switzerland (**Supplement 1 Table S1.1**).” (L 82)

(2.1) Methodology: One possible concern I had regarding the statistical analyses (i.e., GLLVM) is that, given the sampling design and grassland site selection, I think a hierarchical linear regression model would be more appropriate. Essentially, the effects of eco-

These are relevant comments; they allow us to clarify the structure and analyses of the manuscript.

1) The first point raised here concerns the effects of harvest type and eco-scheme and their relation to production system. While it is correct that the study used a blocking structure in the sampling design to ensure an equal number of plots for

scheme and harvest type (along with their two-way interactions) are nested within the production system.

(2.2) In addition, given the paired sites of organic vs. non-organic, a blocking factor needs to be included in the model for proper testing and comparisons.

(2.3) In addition, I would also suspect that other factors such as climatic, vegetation, and soil variables (e.g., soil type, texture, nutrient status, grassland diversity and composition) could also affect ecosystem functions and services. In your models of GLLVM and SEM, have you thought about incorporating and teasing apart any potential confounding factors from climate, soil and vegetation variables, or extent to which the effects of management are manifested through these additional important factors?

the organic and non-organic production system, this does not mean that a statistical analysis has to be performed in a nested way. A nested model would be appropriate where effects of one (treatment) factor must be assumed to differ inherently depending on another (treatment) factor, such as in the analysis of invasive plants in remote continents or countries. In this illustrative example, the invasive plants on different continents must be assumed to belong to different species, which is generally so by observation, due to adaptation to a specific environment and/or phylogenetic reasons. Due to this, also their specific growth performance is usually very different among continents. As a consequence, a reasonable null hypothesis to an interaction between continent x invasive species does not make sense, and responses of invasive (versus e.g., non-invasive) plants have to be analysed nested within continents.

The situation in our study is clearly different, as there is no reason to assume that the effects of eco-scheme extensive and harvest type should inherently differ between the two production systems. This means that a reasonable null for the interaction between eco-scheme/harvest type and production system can be stated, namely that the two do not interact and an eco-scheme or harvest type effect is similar across production systems. This can be tested and we have done so. Notably, the data reveal no interactions of eco-scheme and harvest type with production system, which is not a self-fulfilling outcome of an assumption (compare considerably increased AICc values in (new) **Table S1.2 in the Supplementary Material** for models that included these interactions). Our data show that effects of eco-scheme and harvest type were equivalent across production systems.

Taken together, while our design has a blocking structure, the analysis must be done in a crossed way. To clarify this, we adjusted the description of the plot selection in **Supplementary material S2.2**, L 93. We further removed the part of Figure 1 showing the number of plots per combination of management aspects as this figure was misleading regarding the design and the applied analyses. Because a graphical representation of the data structure would require a 3D figure, the

number of plots per management combination are now referred to as given in **Table S1.1** in the Supplementary Material.

2) With respect to the second point raised concerning the paired sites of organic vs. non organic, we agree that it is appropriate to include a blocking factor (with single levels to include all plots within a pair of organic and non-organic farms). To do this, we added a respective random term “farm pair” to the GLLVM. It turned out that the addition of this factor explained next to no additional variance (equivalent likelihood with and without this random term), meaning that the responses of plots within a farm pair were not correlated to each other. The estimates and standard errors of the fixed effects remained unchanged. This can be explained in that any given farm pair was not very far away from the next farm pair, and plots were generally well distributed across farm pairs and the study region. Thus, the effect of farm pair was in the end negligible. Yet, to avoid any further issue on this aspect, we have updated the models to reveal this directly to readers.

We added this analysis to the methods section: “We included a random factor “farm pair”, with a level each to include all plots within a pair, to account for the blocking structure regarding pairs of organic/non-organic farms in the sampling design.” (L 566)

3) Concerning the last point, the question about climatic, vegetation and soil variables, we carefully considered this suggestion and adjusted the analyses accordingly. We now account for topographic and soil characteristics, and along with these also the local climate, which in the study region depends almost exclusively on the topography (elevation, slope, exposition).

We did not correct for differences in vegetation among the grassland types because these are essentially the result of the management practices studied. Thus, effects of management on ES could not adequately be identified, as

management and plant communities are inherently linked. In addition, because information about the vegetation is included in some of the ES-indicators (plant richness, edible plants, nectar provision), vegetation parameters should not be modelled as predictors to ensure independence between predictors and responses in the analyses.

More specifically, we have now integrated the variables elevation, inclination, and northness (topographic variables that also inform about climatic conditions) as well as pH and sand content (soil variables) into the GLLVM models. We used forward selection and the AICc to find the most parsimonious model that included the three management aspects and the most important environmental variables. The finally selected model included pH, sand content, and elevation, with which we replaced the old GLLVM model in Figure 3. We also added a Table to the Supplementary material (**Supplement 1, Table S1.2**) showing the AICc of the different models that were run. This information allows readers to quickly assess the relative importance of the environmental drivers in this analysis. In addition, this table now reveals that – compared to the previous version – the most parsimonious model does not include any interactions among the three main management aspects studied, which is one of the updated main conclusions of our study (see comments above and below)

Interestingly, in the finally selected, most parsimonious model, which included the three main effects of the management aspects and the three environmental variables, the effects of the management aspects remained almost unaltered compared with the initial model that did not take into account the three environmental variables. This indicates that the three (new) environmental variables explained additional variation in the data but that the effects due to management in our dataset are robust with regard to the environmental gradient. Only the impact of Eco-scheme on microbial carbon biomass became not significant in the updated model, this effect being very close to the significance

threshold in the initial model. We added a Figure showing the model without environment to **Supplement 1, Figure S1.4**, so readers can compare themselves.

Because environmental factors did hardly change any of the effects of management aspects in the GLLVM, we decided against adding the environmental variables to the SEM. This is because we are, for this study, i) mainly interested in the effects of management and ii) of the opinion that the structure of the SEMs will become too complex and distracting if we also add environmental variables. However, we now show that the topography can influence farmers' decision on how to manage the grassland, and thus have added northness to the initial SEM together with elevation and inclination. Doing so, we are also consistent with the GLLVM, which had the three topographical variables tested. Only inclination showed significant effects though, and therefore the results of the SEMs remained unchanged.

As a response to integrating the environmental variables to the GLLVM analysis, we also revised the Data analysis section (L 568 ff).

“The most parsimonious model was identified in the following way: we first ran a simple model including only the main effects of the three management aspects. Then we used forward selection under the second-order Akaike Information Criterion (AICc)89 to determine, which of the five environmental variables (pH, sand content, elevation, inclination, and northness; each transformed to standard deviation scale) resulted in a most parsimonious model. Then, we ran this resulting model including any of the two-way interactions between the three management aspects (see Supplement 1, Table S1.2). It turned out that a model with main effects and three of the environmental variables was most parsimonious (AICc = -2271.3). Given this outcome, we present the main effects in the results section and the effects of the environmental variables in Supplement 1, Figure S1.2. Sand content and pH was measured from the soil samples taken in June 2020. Soil fractions were measured with a SP 2000 Robotic Clay Fraction Analyzer (Skalar Analytical B.V), and soil pH was measured potentiometrically from a water suspension of soil. The

elevation of the plots was derived from a Digital Elevation Model (DEM) of the Copernicus Land Monitoring Service of the European Environment Agency 90 at a resolution of 25 m. From the DEM, northness, representing the orientation of the raster cell to the north, with +1 indicating north, and -1 south, was calculated. In QGIS.org, aspect of the land in radians, and subsequently the cosine of this grid was computed to provide the northness. The inclination of each plot was assessed using the cell phone appliance Clinometer plaincodeTM.

Second, to gain insights on the drivers of the observed effects of the three management aspects on the ES-indicators, structural equation models (SEMs) were calculated in the package lavaan 91. In a first step, the effects of the topographic variables elevation, northness and inclination were tested on Eco-scheme and Harvest type.”

And in the results:

“The main effects and interactions of the three management aspects on the 22 ES-indicators were evaluated with generalized linear latent variable models (GLLVM 32) which revealed that Eco-scheme had the strongest influence on the ES-indicators, while interactions between the management aspects were generally not relevant to explain ES-indicators (Figure 3, Supplement 1, Table S1.2).” (L 119)

“The finally selected GLLVM also included three environmental co-variables to account for potential confounding of the environment with the management aspects (see Supplement 1, Table S1.2 for the model selection summary). Here, soil pH, sand content, and elevation affecting six, seven, and eight ES-indicators respectively (see Supplement 1, Figure S1.2 for the coefficient plots, and Supplement 1, Figure S1.3 for the correlations among all tested environmental variables). Noteworthy, although the environmental variables explained some variation in the data, their addition to the GLLVM had only a marginal impact on the effects of the management aspects compared with a model without environmental variables (see Figure S1.4, in Supplement 1). This is because the management

aspects were the main drivers of plot-scale responses and well distributed across the environmental gradient of the study region.” (L 136)

“Elevation and northness were also tested but removed from the SEM because they did not significantly affect any of the management aspects.” (L 154)

And in the discussion:

“Surprisingly, we did not observe interacting effects of the management aspects on ES to play an important role for multifunctionality. This underlines the relevance of each single management aspect considered separately and the option to freely combine these to achieve the desired set of ES. Mechanistically, land-use intensity was found to be the key driver of single ES and related multifunctionality, as well as of the trade-offs observed between provisioning and especially cultural ES. Thus, the impact of the three management aspects on plot-scale ES-multifunctionality was mediated by restricting land-use intensity, albeit to very different degrees, as will be discussed in the following. » (L 268)

“In line with the positive effect of inclination on the uptake of Eco-scheme extensive management in this study, these indirect effects acting via site history such as long-term extensive management, were suggested to play a crucial role for high plant diversity in permanent grasslands in Switzerland ⁴⁷.” (L 342)

We changed Figure 3 by showing the effects of the three management aspects from the GLLVM including environment and show the coefficient plots for the environmental variables in **Figure S1.2** in Supplement 1. To show the relationships between the environmental variables included in this study, we also added a figure showing Pearson correlations and scatterplots in **Figure S1.3** in Supplement 1. And, as already stated above, we added an AICc table for the different GLLVMs we computed in **Supplement 1, Table S1.2**.

Also, we adapted **Figure 4** with the SEMs, as Microbial biomass was not affected by Eco-scheme anymore after including the environmental variables.

(3) Results presentation: Given the design of the study, the most intriguing part of the results that remains less well emphasized and represented is the interactions among these different management types (examples as shown in Fig. S1.3), in particular for the ES multifunctionality. Because these management practices are often implemented together (as also stated by the authors), the more meaningful and potentially management-relevant findings would be how the combination of management would affect/alter each category of ES, as well as ES multifunctionality.	Thanks to highlighting this strength of our study. We have changed the text according to this comment and now highlight the novelty of considering interactions among the management practices and the lack of significance in the statistical analysis several times within the entire manuscript, including in the abstract. We agree that this aspect makes our work truly unique and increases its applicability to plot- and landscape-scale grassland management for ecosystem services. In the originally submitted version of the manuscript, we had plots showing the interactions between harvest type and Eco-scheme, because the model including this interaction was the most parsimonious (i.e., it had a lower AICc than the model without interactions, see also our newly added Table displaying the AICc values in Supplementary material S1.2, model 4). However, after adding the environmental co-variates to the GLLVM model, none of the models with interactions among management aspects turned out to be more parsimonious than the finally selected one with three environmental variables, which means that the environmental co-variates explained the major part of the variation originally modelled by the harvest type × Eco-scheme extensive interaction. This outcome emphasizes that the main effects of management have the major impact on the ES studied.
(4) Discussion and Interpretation: Much of the current form of discussion relies on re-iteration the patterns from the Result section, without providing much interpretation on what factors or processes would mechanistically drive revealed management effects. Please see examples of my detailed comments below where additional, more in-depth interpretations would	We agree that more in-depth discussions and interpretations of our findings would be valuable additions, and thus changed the text as suggested. Thus, we added passages in the discussion section, for example in L 286 and L 333 (see our answer to the detailed comment below for more details).

be helpful, especially regarding effects of eco-scheme and harvest type).	
--	--

Detailed comments:	Authors comment
L6: consider replacing “either using...” with “harvest type” which has been used in the rest of the manuscript.	Good idea, we adjusted it in Line 7 following.
L33: Why temperate grasslands and their ES rely on grazing or mowing? How about grasslands in other biomes? Grasslands can also provide many ES without moving or grazing. It needs to be clarified.	We meant to say that in many temperate regions (e.g., most parts of Europe), grasslands would be encroached by shrubs and trees and thus not exist without management or grazing activities by large herbivores. We adjusted the sentence to clarify, and it now reads: “In many temperate regions, grasslands and their ES rely on either regular grazing by animals or mowing, as they would otherwise be encroached by shrubs and trees” (L 33)
L38: This statement is largely overstated. There has been a large number of papers focusing on grassland management on ecosystem multifunctionality, either individual land management or multiple practices and across different biomes, for example, Neyret et al. 2021; 2023; Allen et al. 2015; Wang et al. 2019; Schils et al.2022, just to name a few.	We agree, there has been a lot of previous work and we now carefully mention (cite) all these studies. Yet, we would like to point out that that Neyret et al. 2021 and Allan et al. 2015 use the same set of grassland plots (from the biodiversity exploratories project) and Schils et al. 2022 is a meta-analysis with data originating from many different Studies (not all ES are assessed comprehensively from one region but only ordinal changes in services per experiment or study were analysed). This is why we are certain our study is unique, especially since we assessed interactions of the management effects, which has, to our knowledge, hardly been investigated. However, we rephrased the novelty and significance of our study, also in relation to previous work: “However, given the many potentially interacting aspects that shape agricultural grassland management (i.e., mowing versus grazing, different fertilization levels etc.), it has not yet been investigated how management intensity in concert with other key aspects of grassland management affect a broader range of ES and associated multifunctionality.” (L 45)

L45: typo? “1farmer’s choice”	Typo corrected, thanks
L60: “yes versus no” seems not necessary.	We finally decided to keep this terminology to follow the same logic that the other management aspects have (also always two levels, organic – non-organic, pasture – meadow). We further kept “yes and no” here to have short words to refer to the design parameters of the study. Alternatives such as “eco-scheme implemented” versus “not implemented” would have been very unhandy to be used in the text.
L65-66: similar to the comment above, this is likely an overstated argument. There has been quite a lot of studies that have investigated consequences of land-use intensity in grassland multifunctionality	In line with what we responded to the previous comment, we rephrased this passage to take into account that there has already been research in this area. It now reads: “Yet, studies assessing how the effect of extensive management on the simultaneous supply of multiple ES interacts with, e.g., the harvest type, are needed to understanding trade-offs and synergies in ES provision linked to this widespread policy tool ¹³ .” (L 70)
L172-173: Any idea of the relative importance (or proportion of variance explained) between direct vs. indirect effects?	Regarding direct and indirect effects, we think the figures gives already a very good and easy-to-read overview on the relative importance of the effects. We feel that further aggregating these effects would likely lead to an oversimplifying due to the many different drivers involved and because we do not aim at maximizing explained variance in the models but show the respective effects of the three management practices. Moreover, we chose not to include this to not overload the - anyway complex - figure 4. However, the reader can get a good idea as to whether indirect or direct effects play a larger role for each of the ES-indicators (figure 4b). For example, Aesthetics is mainly dominated by direct, and plant richness by indirect effects.
L273: Given the interpretation of the results, this statement may need to be limited to the areas (e.g.,	When rephrasing the text, also because of the other reviewers’ comments, this sentence was deleted to improve the logical flow.

Swiss grasslands) where the background fertilization/pesticide use is much lower for non-organic farming grasslands. Otherwise, broad argument as such could be potentially misleading.	
L302: Please provide explanations on why certain ES (such as weed control and N fixation) are promoted by land-use intensity?	Thank you for pointing this out. We added an interpretation of these results to the discussion: “In the latter study, these ES-indicators belonged to a group of ecosystem functions and processes that depend on high nutrient input such as microbial biomass, nitrification, and earthworm abundances ⁴³⁻⁴⁵. Alternatively, they can be driven by further aspects of intensive management like early harvest dates (e.g., less invertebrate herbivory ⁴⁶) and weeding activities, as farmers seem to accept more (potentially ecologically valuable) weeds in extensively managed grasslands, while focusing weed management on non-Eco-scheme (intensive) grasslands. In the case of N₂ fixation, the indicator used was strongly driven by biomass production, explaining the positive influence of non-Eco-scheme management and related fertilizer inputs.” (L 330) We also added some more explanations for other indicators, in L 284 f: “This lower land-use intensity of organic grassland farming, which is connected to the ban of synthetic fertilizers and lower limits for organic fertilization, has also been observed in other contexts ^{33,34} and could directly be responsible for the observed lower N₂O emissions from organic grasslands. As lower N fertilization also relates to lower P input, this might also explain the higher abundance of AM fungi in organic grassland soils, indicating lower soil P availability compared to intensively fertilized conventional soils”
L325-326: Sorry if I am confused here. I thought that based on the methods, the mean LRR was calculated using the mean values of LRR for each indicator, which	It is correct that we use the mean LRR as a measure for differences in ES-multifunctionality, and we calculated first the LRR for every indicator separately for each comparison we wish to do (e.g., Eco-scheme extensive yes vs. no) (see

essentially occurs to me as a simple average approach? I'd appreciate if you can further clarify how the MLRR approach differs from and helps address the black box regarding the contributions of single indicators.

descriptions on L 622 f and more details in the Supplement). Then, we averaged the LRRs representing the same CICES-ES and give the justification for this step on L 631: "Doing so, ES-indicators informing about different aspects of one ES were down-weighted to avoid over-representation of one CICES-ES over the others (...) We chose to use the CICES-ES framework as it is widely used in science and practice and thus allows for a better comparability of the results to other work." The pooling ES-indicators to CICES-ES is mathematically equivalent to taking an average across *all* LRRs and weighting ES-indicators by 1 divided by the number of respective indicators per CICES-ES. For example, 3 ES-indicators contribute to the CICES-ES "Pest control" (**Figure 1**). Our step of aggregating these 3 ES to "Pest control" is equivalent to multiplying these LRRs by 1/3, if one would take a simple average across *all* LRRs to receive the MLRR.

Given this, the MLRR approach is quite different from the so-called averaging approach, which calculates an average MF-value from all indicators per plot. In this latter approach, all information about the single indicators is essentially lost in this step. For example, ES-indicators can have (equal) opposite values or equivalent values, yet both cases will result in the same average metric. Opposite effects cannot even be evaluated after averaging (and this is where the "black box" comes in), which makes the interpretation of a simple plot average as a measure of multifunctionality challenging if not impossible. In our approach, the LRRs representing a CICES-ES were plotted (**Figure 5**), and it becomes clear that this is where the "black box" is avoided, as we can still see which ES profits and which loses under specific management.

Regarding the general advantages of the MLRR approach, we have listed some of these in the Supplement of the originally submitted manuscript. To emphasize this aspect, as requested by the reviewer, we have shifted a modified version of this previous paragraph to the data analysis section in the revised manuscript. See L 642 ff.

	“To support our results on overall multifunctionality based on the MLRR, we evaluated the effects of the management aspects (organic vs. non-organic, Eco-scheme extensive yes vs. no, and pasture vs. meadow) on multifunctionality using two further methods. First, we calculated a mean multifunctionality index using the estimates of the GLLVM as shown in Figure 3, and second, we calculated multifunctionality using the averaging approach. Using these two alternative approaches, we found the multifunctionality results to be highly similar to the MLRR. Based on these similar outcomes from two alternative methods, we conclude that our results for the MLRR are not only well interpretable but also highly reliable. See new paragraph in Supplement 2.1 “Calculating the mean log response ratio...” for details to these calculations.”
L330-332: Please provide any possible mechanisms on the diverging effects of harvest types.	In the revised manuscript, we provide mechanisms for these effects: “This, together with effects of trampling and unselective mowing versus highly selective grazing differently shaping plant communities and their traits ⁵⁷, likely explain the observed differences in ES observed between the two harvest types. For example, the negative effect of grazing on earthworms was most probably due to soil compaction by trampling livestock ⁵⁸, which has also been shown to potentially reduce yields ⁵⁹, in line with our study. On the other hand, trampling and the selective grazing by livestock will have increased plant richness, in addition to the slightly decreased fertilizer use in pastures.” (L 370)
L382: A study map of the grassland ecosystems, as well as locations/(possibly photos) of the study plots would be very informative – e.g., to demonstrate spatial gradient of these plots.	We agree, a study map with the location of the plots would be helpful. Yet, we cannot show and share the exact location of the plots, so as to protect the privacy rights of the participating farmers. However, we added a map to the supplementary material showing the study region and how many farms and plots were sampled from which biogeographic area of the canton (Figure S1.1 in Supplement 1).

L452-474: Some sampling details seem to be missing. For example, how many 2x2 m quadrats per plot were used for sampling vascular plant species richness, and are they randomly placed in each plot or along the transect? Were the edible plant and nectar plant also sampled from the 2x2 m quadrats, and if not, how were they sampled? For leave damages, how were the leaves sampled and collected, and how much were sampled for each plots?	That is correct. We now give the requested information in the main text, as we agree with the Reviewer that this information should be clear without referring to the Supplement (L 516, 518, and 521). Generally, we did not provide all the methodological details in the main text but have provided all the sampling and other methodological information in the Supplementary material S2.3. Due to restrictions in article length, we cannot address every detail in the main text, given that we have measured so many ES-indicators.
L483: Please specify what tools used for quantifying potential N leaching.	We agree that this could be improved. We clarified it by adding more information. It now reads: “Information on N inputs was also used to estimate potential nitrate leaching using a tool developed by the UK Environment Agency accounting for fertilizer N and animal excreta as sources for nitrate leaching⁸⁹.” (L 547)
L497: Please specify what are the two specific latent variables in the GLLVM and the justifications.	The latent variables within a generalized linear latent variable model account for the correlation of responses, which must be done in all multivariate analyses to receive a correct inference on the (fixed) estimates. In the classical multivariate approaches, a full co-variance matrix is imposed on the multivariate residual distribution. However, there are limits to this as the number of covariance terms increases non-linearly with the number of response variables, which quickly leads to convergence problems in model fitting. The GLLVM approach suggested by Niku et al. (2019) solves this challenge by using latent variables so that the number of parameters for covariances scales linearly with the number of responses. To provide a basic meaning, we have added the sentence in L 564: “The latent variables can be thought of as ordination scores, capturing the main axes of covariation of responses after controlling for observed predictors” (adapted from Niku et al. 2019).” For more technical details, we refer to Niku et al. 2019.

	We added a justification for using 2 latent variables to the main text: “All regressions implied a Gaussian link and two latent variables, as models with more than two latent variables did often not converge. Where models converged, more than two latent variables did not change the fixed estimates and their standard errors.” (L 562)
L535: How the bootstrap was performed to calculate MLRR can be further clarified; for example, in Supplementary S2.1, it mentioned to bootstrap 1000 times from the replicated plots (i.e., 86 plots total). Can you specify what percentage of samples was randomly selected from each iteration of bootstrapping?	All available data is used for bootstrapping. We followed the basic method where the same number of units as available in the original dataset is sampled with replacement in each iteration. This is now clarified in the Supplementary material S2.1: “Thus, we sampled 1000 times with full replacement from the replicate plots within the 8 groups of management aspects and calculated the LRRs as outlined above to receive a distribution of MLRRs for each comparison” (L 38)
Fig. 1b: For the graphics of production system – organic, do you mean a “x” (instead of check mark) on the synthetic pesticides icon?	That is a very helpful comment, as we have noticed that the respective part of the figure was indeed confusing. The icon was in fact not indicating synthetic pesticides, but mineral fertilizer. We now have 3 correct icons, one for synthetic pesticides, one for mineral fertilizer and one for organic fertilizer. Thus, we adapted Figure 1 and think it is much more logical and clearer now.

References cited in the comment

Neyret, M., Fischer, M., Allan, E., Hölzel, N., Klaus, V.H., Kleinebecker, T., Krauss, J., Le Provost, G., Peter, S., Schenk, N. and Simons, N.K., 2021. Assessing the impact of grassland management on landscape multifunctionality. *Ecosystem services*, 52, p.101366.

Neyret, M., Peter, S., Le Provost, G., Boch, S., Boesing, A.L., Bullock, J.M., Hölzel, N., Klaus, V.H., Kleinebecker, T., Krauss, J. and Müller, J., 2023. Landscape management strategies for multifunctionality and social equity. *Nature Sustainability*, pp.1-13.

Allan, E., Manning, P., Alt, F., Binkenstein, J., Blaser, S., Blüthgen, N., Böhm, S., Grassein, F., Hölzel, N., Klaus, V.H. and Kleinebecker, T., 2015. Land use intensification alters ecosystem multifunctionality via loss of biodiversity and changes to functional composition. *Ecology letters*, 18(8), pp.834-843.

Wang, L., Delgado-Baquerizo, M., Wang, D., Isbell, F., Liu, J., Feng, C., Liu, J., Zhong, Z., Zhu, H., Yuan, X. and Chang, Q., 2019. Diversifying livestock promotes multidiversity and multifunctionality in managed grasslands. *Proceedings of the National Academy of Sciences*, 116(13), pp.6187-6192.

Schils, R.L., Bufe, C., Rhymer, C.M., Francksen, R.M., Klaus, V.H., Abdalla, M., Milazzo, F., Lellei-Kovács, E., ten Berge, H., Bertora, C. and Chodkiewicz, A., 2022. Permanent grasslands in Europe: Land use change and intensification decrease their multifunctionality. *Agriculture, Ecosystems & Environment*, 330, p.107891.

Reviewer #2 (Remarks to the Author):

Summary:

The authors evaluated the impacts of three grassland management practices (i.e., organic farming, eco-scheme extensive management, and grazing vs mowing) on the ecosystem services (ES) and their multifunctionality at the plot level in the Canton of Solothurn in the northwest of Switzerland. The study selected 22 ES indicators covering provisioning, supporting/regulating, and cultural services, and used multivariate regression and structure equation modeling for statistical analyses. The main findings were: (1) organic farming did not improve ES multifunctionality as expected; (2) eco-scheme extensive management improved the overall ES multifunctionality most substantially – increasing a number of regulating and cultural services at the expense of provisioning services; (3) the different management practices affected ES mainly by changing land use intensity.

The topic of the study, ecosystem service multifunctionality, is certainly important to sustainable ecosystem management, sustainable land planning, and landscape/regional sustainability. The dataset looks unique and interesting. The statistical methods seem appropriate for addressing the research questions (although they were not explicitly stated in the manuscript).

Response: Thank you for assessing our manuscript, and for this positive evaluation of our work.

The findings about the effects of different grassland management practices on individual ES and their tradeoffs generally corroborated those from many previous studies in different parts of the world. So, they are not new, but just similar findings from a Swiss region. Nevertheless, the three main findings are interesting and may potentially have important implications for grassland management in the study region.

Response: While we are happy to read that you find the conclusions drawn from our study important and relevant, we politely disagree on the statement saying that the assessment of the different grassland management practices is not really new. Yes, there have been studies looking into some aspects related to grassland management practices and have assessed single or few ecosystem services. Especially the effect of management intensity has priorly been looked, as we state in our introduction (see also our responses to the comments by reviewer 1). Yet, importantly, the scale of our study, the large set of services assessed, and especially the fact that we, other than previous studies, also consider all possible combinations of the management aspects make our study outstanding from previous work. We agree this novelty might not have been very obvious in the former version of the text and, thus, we improved this in the revised introduction and discussion (e.g., L 45 or 70).

However, I am afraid that some of the major conclusions and recommendations were not justified or supported by their results. The main reasons for this are some fundamental problems with the conceptualization and analysis framework of the study. Let me summarize them briefly below:

Response: Thanks for pointing this out. In the former version of our manuscript, it was indeed not very obvious, for example at which scale our study was conducted and what are the suggestions we make for managing ecosystem services and multifunctionality at the landscape scale. This lack of clarity will have led to the confusion that was obviously caused regarding conceptualisation and results. We have intensively worked on the paper to improve these aspects, with an emphasis on explaining the conceptualisation in more detail and showing more explicitly why our study is key for managing and enhancing multifunctionality at the landscape scale.

You can find our answers to your single comments below. In quotes from the manuscript text, new or altered passages are highlighted in yellow.

Reviewer's comments	Author comments
1. All analysis was conducted at the plot level (i.e., "grassland parcels"), but the focus of the study was about	Thank you for highlighting this confusion, which might to a large degree originate from the initial text not always specifying whether we refer to plot- or

ES-multifunctionality. This seems a major reason why the results were not really addressing the research objectives. At the plot or site level, ES tradeoffs are inevitable and easily understandable because each plot (or site) may be used mainly for a particular purpose (or one primary ES or a group of ES). Thus, ecosystem service multifunctionality needs to be studied and achieved at the landscape (as well as regional) scales. This seems the dominant view in the current literature in landscape multifunctionality and landscape sustainability. Indeed, it is a key tenet of the emerging research field, landscape sustainability science (<https://link.springer.com/article/10.1007/s10980-013-9894-9>). Unfortunately, although the manuscript mentioned “landscape-scale sustainability” once in passing, it’s not clear at all how the plot-level ES multifunctionality is related or relevant to landscape sustainability.

2. Related to the above, most of the results and conclusions are confusing and, sometimes, nonsensical, because there was not an explicit spatial scale attached to it (e.g., plot or landscape). Diversifying ES at a “plot” level sounds unpractical as a plot is rather small in size. It seems more appropriate and much more informative to conduct the management strategies-ES multifunctionality analysis on multiple scales: from plots/sites to landscapes/regions. This way, the plot-level and landscape level ES multifunctionality can be distinguished and compared. More importantly, the impacts of landscape composition and configuration on ES multifunctionality – a key research

landscape-scale multifunctionality. This has now been improved, as you can see from the following responses. In addition, we fully agree, also based on our findings, that a supply with all ES can only be achieved at the landscape level. However, assessing plot-level multifunctionality is a pre-requisite, the first inevitable step, if landscape-level multifunctionality is to be investigated and improved, as land management always targets the plot level. This results from the fact that landscape-level multifunctionality is influenced a) by the cover of different ecosystems and their *plot-level* services within the landscape and b) the configuration of these ecosystems, their spatial placement, topographically and relative to each other. It was, however, not in the scope of this study to look at both of these aspects, also because of the given limits regarding text length. In lieu thereof, we concentrate on the former aspect of the two, which already relates to landscape-scale management for ES (see below) and can, in further investigations that include spatial analyses, inform spatially explicit assessments of landscape-scale multifunctionality. Thus, although not studying landscape configuration, our findings of management effects on single and multiple ES (i.e., multifunctionality of e.g. all cultural services as shown in Figure 5) are of direct and strong relevance for (i) managing specific ES that are demanded from a given landscape, and (ii) increasing ES-multifunctionality at landscape scale.

We have taken this misunderstanding of the relevance of our study for landscape-scale multifunctionality very seriously when revising the paper. To remove this misunderstanding, we now elaborate in the introduction on the need to overcome inevitable plot-scale trade-offs between services at the landscape level. We furthermore argue in the introduction that informing land management at the plot scale is still needed to enhance the provision of (sets of) ecosystem services that are short in supply. We additionally address the different meaning of plot-scale and landscape scale. (L 40 f).

“Many of these agricultural and agri-environmental regulations target plot-scale grassland management, resulting in different plant communities and delivering

topic in landscape sustainability research – can be investigated.

different sets of ES and different levels of plot-scale multifunctionality¹². The latter can inform about how a broad set of ES is affected by a specific management practice. Knowledge on plot-scale effects of management on ES is further required to achieve multifunctionality on the landscape scale, resolving inevitable trade-off between services at the plot scale^{15,16}”

In the discussion, we directly refer at several places to how our results can be used to increase landscape-scale multifunctionality (directly in the beginning of the discussion L 260 ... “We found the three major grassland management aspects studied to differently affect single ES and ES-multifunctionality at the plot-level. This insight in effects of grassland management, as shaped by agricultural policies and local decision-making, on single ES and plot-scale ES-multifunctionality is required to take informed action aiming to maintain and enhance landscape-scale multifunctionality and meeting societal needs that go beyond food production”

and L 316: “Our study provides evidence for a strong positive effect of the **Eco-scheme** ‘extensive grassland management’ on ES-multifunctionality and especially cultural ES-multifunctionality at the plot scale. This finding is of particular relevance for managing such ES in landscapes dominated by intensive grassland management. “

and L 377: “The fact that Harvest type significantly influenced many single ES and impacted ES multifunctionality shows that this aspect of grassland management could indeed be an impactful lever in adjusting ES supply in a given area, depicting a relatively easy-to-implement tool to enhance cultural ES and landscape-scale ES-multifunctionality. Our study implies that landscapes dominated by grassland-based livestock systems relying on outdoor grazing will deliver a set of ecosystem services less supported by livestock systems with all-year indoor feeding.

The inevitable trade-offs we observed among different sets of ES lead to the conclusion that finding a one-type-fits-all grassland management regime is impossible, and that multifunctionality needs to be finally achieved at the landscape scale by allocating different management to different areas within the landscape (i.e., spatial segregation of ES production)¹¹. Yet, only a well-informed use of different management approaches, as provided by our study, can optimize ES-multifunctionality as desired by the local stakeholders ¹⁵.”

Generally, throughout the text, we now always clearly indicate whether we speak about plot- or landscape-scale multifunctionality.

In addition, we wish to note that plot-scale multifunctionality actually is an established concept, which does not claim to overcome trade-offs among services but which can inform about the effects of plot-scale management on a broad range on ecosystem services. There is a large body of literature in high-impact journals that follows this concept of plot-scale multifunctionality, e.g., in grasslands: Allan et al. 2015 (in *Ecol Lett*) and Le Provost et al. 2023 (in *Nature Ecol Evol*); in drylands, for example: Maestre et al. 2012 (in *Science*) and Valencia et al. 2015 (*New Phytol*); in forests, for example: van der Plas et al. 2016 (in *Nat Commun*) and van der Plas et al. 2018 (*Ecol Lett*); and those studies linking biodiversity to multifunctionality such as Gamfeldt & Roger 2017 (in *Nature Ecol Evol*), Soliveres et al. 2016 (in *Nature*) and Hector & Bagchi 2007 (in *Nature*). See also the conceptual paper by Manning et al. 2019 (in *Nature Ecol Evol*) and a multi-ecosystem review in Garland et al. 2021 (in *J Ecol*), in which it becomes obvious that drivers of plot-scale multifunctionality are relevant to understand the effects of land management on ecosystem services and the various benefits humans derive from ecosystems. In addition, we might be allowed to mention that assessing multifunctionality at plot-scale was not considered as a critical issue by the other two reviewers that assessed our work. Therefore, we added a sentence in the Results section to elaborate on this: “Despite diverging responses

	of single ES to Eco-scheme and Harvest Type pasture, overall ES-multifunctionality significantly increased with these two management practices, highlighting that grassland management can considerably affect multifunctionally at the plot scale (Figure 5a).” (L 197) Generally, we agree that any kind of multifunctionality measure can be a black box, as it is not obvious which (set of) services decrease or increase in a given setting. Thus, multifunctionality measures have a limited value for actual land-use decisions. Therefore, in this work, we (i) put the main emphasis on the effects of management on single services (see Figures 2 to 4) and (ii) when assessing multifunctionality (Figure 5), we also show results on the three separate ecosystem services categories (provisioning, regulating and cultural), which gives a very comprehensive overview on how management drives different sets of ecosystems services. We decided to leave the title of the paper without the addition of plot- or landscape-scale (to multifunctionality) because both spatial dimensions are being addressed in our paper, with the results also being relevant for both scales. Plot-scale as part of the methodological approach, and landscape-scale to make specific recommendations how our conclusions can inform land management to increase multifunctionality at larger spatial scales. Given all these changes and adjustments, we are certain the paper no longer allows for any confusion regarding the spatial scale we are referring to.
3. Simply counting the number of ES as the measure of multifunctionality may lead to ecologically unwarranted conclusions. For example, ES related to water/soil pollution ought to be emphasized because they are essential to ecosystem functions and human health. Also,	Thanks for bringing this up; this is an important point. We did not count ecosystem services, also not in the multifunctionality metric we use, and it is important to us in our work to clearly show which (sets of) ecosystem services are positively or negatively impacted by the different management practices.

eco-scheme extensive management did not use fertilizers and pesticides and have low levels of land use intensity, and consequently a number of regulating and cultural ES increased, at the expense of provisioning ES. But, can a region be sustainable without local food production?

We understand that the reviewer is concerned because she/he assumes we recommend using only the management aspects that results in highest plot-scale multifunctionality. Yet, this is a misinterpretation. As elaborated above, multifunctionality is only a part of our study, actually only the last step in the chain of analyses, and it helps (on top of all the analyses focussed on ES-indicators and single ES) to find out which management results in highest overall plot-scale ES-provision. This can actually indicate less trade-offs between the services included, and allows for a comparison with management practices that reduces plot-scale multifunctionality. We clarify this in the introduction by adding an explanation saying plot-scale multifunctionality can only be used to assess how many trade-offs in the studied set of services occur in L 40 f:

“Many of these agricultural and agri-environmental regulations target plot-scale grassland management, resulting in different plant communities and delivering different sets of ES and different levels of plot-scale multifunctionality¹². The latter can inform about how a broad set of ES is affected by a specific management practice. Knowledge on plot-scale effects of management on ES is further required to achieve multifunctionality on the landscape scale, resolving inevitable trade-off between services at the plot scale^{15,16}. However, given the many potentially interacting aspects that shape agricultural grassland management (i.e., mowing versus grazing, different fertilization levels etc.), it has not yet been investigated how management intensity in concert with other key aspects of grassland management affect a broader range of ES and associated multifunctionality.”

We do certainly not suggest using only the practice that results in highest local multifunctionality (see comments above regarding the relevance of the landscape scale and how our conclusions help to manage services at this scale), and to make this bullet-proof, we directly state this in the discussion “The higher plot-scale multifunctionality observed should not be mistaken as the overall best way to manage grasslands, but suggests that Eco-scheme extensive

management results in overall less trade-offs among the services studied.” (L 319)

As for all other studies on ES-multifunctionality, we further agree that the outcomes of our study is indeed highly dependent on the ES that are included, and additionally on how they are weighted (see also Allan et al. 2015 and Neyret et al. 2023, cited above and in the main text). This is why (i) we assessed as many ecosystem services from all three categories (provisioning, regulating, cultural) as was possible, and (ii) we chose to use the LRR approach to analyse the data based on the single services, also separately for these three categories: Thus, in Figure 5, it is very transparent and clear, which trade-offs exist, which ES increase and which decrease under a given management scenario. We also further clarified this in the text: “Yet, such issues can be easily detected and put into context by the assessment of single ES indicators and the MLRR approach used here, as the MLLR approach allows for a transparent assessment of the contribution of single ES.” (L 357)

We also added some clarification at the end of the introduction:

“Results of this study can, thus, inform and support improving grassland management and related agri-environmental policies in optimizing grassland ES provision and thus improving the multifunctionality of agricultural landscapes. Insights into the relationships between single practices and ES allow farmers and other decision makers to adapt grassland management to support a specific ES or even ES-multifunctionality at a given site.” (L 96)

Also, we added some clarification to the Discussion section:

“The inevitable trade-offs we observed among different sets of ES lead to the conclusion that finding a one-type-fits-all grassland management regime is impossible. Only a well-informed combination of different management approaches can optimize ES-multifunctionality as desired by the local

	stakeholders⁶³. To do this effectively, the ES demand and priorities of local stakeholders have to be translated into a regional management plan to optimize the ES-provision^{14,60,61}, as the stakeholders' rating of the importance of a given ES differs according to the region and context⁶¹." (L 383)
4. The three management strategies did not seem really "independent" as they claimed because, for example, Eco-scheme had mowing whereas pastures also had mowing (vice versa – meadows also had grazing). Thus, the statistical analysis did not consider these complications. Thus, I suggest that you will deal explicitly with the problem of data interdependence (e.g., considering only pastures without mowing and meadows without grazing).	This comment seems to result from a misunderstanding, which we would like to explain here, and which we now explicitly clarify in the paper to avoid any confusion of future readers. We use a fully crossed factorial design (as explained in detail in the answer to Reviewer 1, Comment 2, above), in which the management strategies are implemented independently from each other. It is correct, that some of the meadows could be subject to a brief autumn grazing, but essentially, they depict a distinct grassland type following the Swiss landscape typology (Nomenklatur Arealstatistik, Federal Statistical Office, 2018), which defines the types of meadows and pastures. This is also how farmers in reality manage their land, as most grasslands are not strictly managed by only grazing or mowing. Thus, both practices are to some degree combined to keep a desirable plant species composition. For example, infrequent cutting in pastures is used to reduce weeds that are not completely grazed by livestock. However, this grassland typology is widely applied in practice and research (e.g., Klaus et al. 2023, Journal of Environmental Management 348: 119416. To fully clarify this, we added a more detailed description of the harvest types meadow and pasture to the methods: "For the Harvest type, we chose the two dominant grassland types occurring in Central Europe. While meadows are predominately mown, with some occasional grazing such as at the end of the growing season, pastures are mainly grazed and rarely cut. This differentiation follows the official typology for Swiss grasslands and was confirmed by farmer interviews (Supplementary material Table S1.1)." (L 459)

	Our study plots are absolutely in accordance with these definitions, as becomes clear in Table S1.1, where the actual management of the plots is summarized. Our selection of plots thus depicts a typical sample of meadows and pastures in a realistic agricultural setting. Thus, we are highly confident that our results are meaningful and robust, and realistically represent the management strategies implemented by the farmers. Thus, there is no need to account for any nesting or independency in the study design involving the three management practices (but see the implementation of the random factor “farm pair”, in response to reviewer 1, which, yet, did not change the results).
5. In the Discussion section, the authors stated: “the ES demand and priorities of local stakeholders have to be combined with plot-level assessments of ES from studies like here in a regional management plan to optimize the ES-provision on the landscape scale ... Thus, ... it is necessary to adopt a wider view that includes farm and landscape scale.” I agree, but optimizing ES multifunctionality at the landscape scale requires the consideration of both composition and configuration (spatial arrangement). The latter was not studied in this study. Then, they concluded: “diversifying grassland management where this is currently rather homogeneous across farms and landscapes would be an important and effective step to increase ES-multifunctionality for sustainable grassland farming.” Well, again, this sounds right, but ES diversification and landscape sustainability require the consideration of not only the spatial pattern of ES, but also their tradeoffs across spatial and temporal scales.	We fully agree with this point, as also discussed in our answer to Comment 1, 2, and 3 above. We have revisited this paragraph and improved the phrasing to enhance clarity (see below). After rephrasing the introduction and discussion, we are now very confident it is now obvious that (i) plot-scale multifunctionality is not describing the only recommended management practice but a situation where trade-offs are reduced, (ii) our conclusions are of direct relevance for landscape-scale multifunctionality, and (iii) we set an emphasis to showing the changes in single ecosystem services to allow understanding all trade-offs observed for a well-informed decision making at field and landscape scale (Figure 2 to 5). Moreover, to highlight the relevance of landscape configuration, we added a paragraph directly before the Conclusions to emphasize this point: “Thus, in addition to our plot-level results, it is necessary to adopt a wider view that includes farm- and landscape-scale drivers, as for example not only landscape composition but also configuration are important for ES-provision. For instance, eutrophication potential depends on hydrological conditions around the site, and

	resources for pollinators might have a higher importance close to insect-pollinated crops ¹¹.” (L 390) We also added a sentence to the Conclusions to point out future research need, which goes exactly in the direction you suggest for addressing landscape-scale multifunctionality in future studies: “Overcoming these trade-offs should receive further attention in future research and practice. Building on plot-level assessments of management effects, such as the one carried out in the present research, investigating landscape-scale ES-multifunctionality considering both landscape composition and configuration, while suggesting new or alternative ways to manage grasslands (e.g., increasing plant diversity of swards ¹²), could have the potential to additionally benefit the portfolio of ES provided by agricultural landscapes. Meanwhile, our plot-level results very clearly suggest that diversifying grassland management where this is currently rather homogeneous across farms and landscapes would be an important and effective first step to increase ES-multifunctionality for sustainable grassland management. (L 426)
6. The effects of the selection of ES indicators on the results have been reported in recent studies, which should be discussed in the context of this study	Exactly, this is a relevant point, which we now directly address in the Discussion, when speaking about the impact of single indicators on cultural ES-multifunctionality: “This case of one indicator strongly driving a measure of multifunctionality, as well as the reasoning that multifunctionality is strongly influenced by the choice of indicators, points to one of the drawbacks of the frequently applied averaging approach, for which the average index operates as black box regarding the contributions of single indicators ^{51,52}. Yet, such issues can be easily detected and put into context by the assessment of single ES indicators and the MLRR approach used here, as the MLLR approach allows for a transparent assessment of the contribution of single ES.” (L 353)

7. The use of the term, “farming”, in this paper may be reconsidered because its narrow definition usually refers to cultivation or crop production.	We agree, terminology can always be understood differently, especially if it concerns broad terms such as farming. In this case, however, the use of “farming” in the title emphasizes the fact that we are dealing with grassland management as implemented by farmers. However, in the remaining text, we avoided the word “farming” and only use “management”. In the title, however, there needs to be farming in the beginning as management is needed for the part following the colon.

Reviewer #3 (Remarks to the Author):

Summary:

Dear Editor,

This manuscript provides a comparison between different management systems for the Swiss Alps, which is potentially representative of European temperate grasslands. The study is based in 86 plots quite evenly distributed between organic and non-organic, extensive and intensive, pastures and meadows. Authors analyse the contribution of each of the 8 possible combinations of management types on 22 ES indicators. The text is well written and the conclusions are supported by the results, which show quite strong evidence for their findings (main differences between production systems are intensive versus extensive and secondarily pasture vs meadow rather than organic vs non-organic). However, given the number of studies currently published on temperate grasslands multifunctionality, the novelty of the approach and findings needs to be better justified. A few points for improvement of the manuscript remain:

Response: Thank you very much for the positive evaluation of our work. We addressed all points raised and are confident this has additionally improved the paper. Thank you!

You can find our answers below. In quotes from the manuscript text, new or altered passages are highlighted in yellow.

Reviewer's comments:	Author comments:
1. The text of the abstract could be improved to better capture readers' attention. For example, it needs a better introduction and justification of the novelty of the study and their findings.	We attempted to re-write the abstract to better capture the novelty of the study: "Human wellbeing depends on strategies to maintain and enhance ecosystem services (ES) provision. Thus, improved multifunctionality of food and feed production is urgently needed. We assessed how employing and combining three widespread aspects of agricultural grassland management, namely i) organic production system, ii) eco-scheme 'extensive management', and iii) harvest type (pasture vs. meadow), can enhance plot-level ES and respective multifunctionality, based on 22 ES-indicators. While organic production system and interactions between management aspects played a minor role, the main effects of eco-scheme and harvest type considerably shaped ES-provision. Moreover, eco-scheme 'extensive management' and harvest type 'pasture' enhanced plot-scale ES-multifunctionality, mostly through facilitating cultural ES at the expense of provisioning ES. Changes in ES occurred via changes in land-use intensity, i.e., fertilizer input and harvest frequency. In conclusion, different options exist to manage grasslands for specific environmental benefits and to combine these options to improve landscape-scale multifunctionality to meet societal ES demand."
2. The term eco-scheme could simply be named extensive/intensive, which would avoid misunderstandings, given that the term eco-scheme is very broad and can imply other aspects too.	We agree that the term eco-scheme is broad and potentially encompasses many aspects. To clarify this throughout the paper, we adjusted our terminology by adding "extensive" to eco-scheme. Therefore, we also changed Figure 1b that illustrates the three management aspects and changed "Eco-scheme" to "Eco-scheme extensive management". Moreover, we wish to avoid speaking of extensive/intensive as a contrast because (i) these terms are not universally defined and (ii) the management on the plots not

	under eco-scheme extensive management ranges from mid-intensive to highly intensive. This means that the management in the non-Eco-scheme plots is quite variable, as it is normal for agricultural landscapes (as you can also see in the standard deviations for the management inputs, in Table S1.1). Therefore, we decided to avoid only speaking of “extensive versus intensive” management.
3. The extensive eco-scheme does not seem to be independent from organic, given that all organic farms are extensive and extensively managed grasslands do not receive any fertilizer in this study. It is not clear how authors account for this interaction.	Thank you for bringing this up. This was due to a phrasing issue, which we eradicated by improving the description of the management aspects studied, and how these relate to each other. The eco-scheme is indeed independent from organic, as (i) both organic and non-organic farms can participate in it and (ii) organic grassland management is not necessarily more extensive than non-organic management. We also adjusted Figure 1b to clarify this. Mineral nitrogen is prohibited on organic grasslands, but farmers could compensate via higher amounts of organic fertilizer. Also, the lack of pesticide application can translate to more mechanical interventions, which also severely impact the soil and ecosystem functioning. Thus, organic management is independent from “management intensity”, which is mainly defined as fertilizer amount, grazing intensity, and cutting frequency. On extensive grasslands, both organic and non-organic farming must follow the same regulations, banning any kind of fertilizer and restricting earliest cutting date. We are confident that the changes implemented fully clarify this aspect in the current version of the paper.
4. Somehow the text seems to avoid supporting organic farming practices, as not having additional positive effect. However, these results can also be interpreted as organic farming not having a negative effect on provisioning ES, which is often the key criticism of this management system. I would suggest to hence present these results in a more neutral tone.	We did not intent to judge organic farming beyond the results we obtained in our study and thus carefully checked and improved our presentation of the effects of organic farming. This led to several changes in the revised text, mostly because we moderated the conclusions to some degree. Indeed, a wide yield gap is often used in discussions against organic farming, while our paper tells a somewhat different story in this regard. However, in line with previous studies, the decrease in the ES-indicator “biomass production” under organic

management is considerable, even if not significant due to large variability. Thus, we find it fair to mention this obvious trend.

Therefore, we rephrased the text and also highlight the positive aspects that organic management provided in our assessment:

“Organic grassland farming improved two out of the 22 ES-indicators and did not significantly improve overall ES-multifunctionality. However, importantly, organic farming did not have any significant negative effects on ES-indicators or ES-multifunctionality, but showed a tendency towards lower biomass production. The overall small effect of Production system is most likely due to rather small differences in management intensity between organic and non-organic grasslands.” (L 277)

“Yet, despite lower fertilization intensity, there was no statistically significant negative effect of organic management on provisioning ES-multifunctionality, which highlights a strength of organic management in this respect.” (L 289)

“The very small marginal benefits of organic management observed here close the research gap concerning the portfolio of ES supplied by organic grasslands” (L 295)

“Regarding organic management, we observed very weak effects of this management aspect on plot-level ES, but organic management system could have further effects on ES at farm and landscape scales.” (L 396)

Also, we added a clarification in the methods section referring to the exact rules applying for organic grasslands in Switzerland:

“In Switzerland, the maximum allowed amount of organic fertilizers applied per year and hectare is somewhat lower than for non-organic grassland farming (135 vs. 162 kg available N for all non-Eco-scheme grasslands of a farm at low elevations⁶⁸).” (L 446)

5. The conclusion section would benefit for more concrete statements. For example, instead of “regulate land use intensity”, authors could point out to which levels of land use intensity should be recommended, and how to deal with trade-offs.

Thank you for this valuable suggestion, with which we agree. We therefore added more concrete statements to make the manuscript more mechanistic and thus also more applied.

Thus, we amended the text in several places in the discussion and conclusion sections, and framed detailed recommendations on how our research results can be used to manage multifunctionality, especially also to translate into landscape-scale management (see comments of reviewer 2).

For example...:

Our study provides evidence for a strong positive effect of the **Eco-scheme** ‘extensive grassland management’ on ES-multifunctionality and especially cultural ES-multifunctionality at the plot scale. This finding is of particular relevance for managing such ES in landscapes dominated by intensive grassland management.” (L 316)

“The fact that Harvest type significantly influenced many single ES and impacted ES multifunctionality shows that this aspect of grassland management could indeed be an impactful lever in adjusting ES supply in a given area, depicting a relatively easy-to-implement tool to enhance cultural ES and landscape-scale ES-multifunctionality.” (L 377)

“The inevitable trade-offs we observed among different sets of ES lead to the conclusion that finding a one-type-fits-all grassland management regime is impossible, and that multifunctionality needs to be finally achieved at the landscape scale by allocating different management to different areas within the landscape (i.e., spatial segregation of ES production)¹¹. » (L 383)

" As no strong interacting effects of the management practices studied were observed, these practices can be freely combined to achieve the desired set of services. This way, our plot-level outcomes can directly translate into action for landscape-scale management for ES-multifunctionality.” (L 417)

	“Overcoming these trade-offs should receive further attention in future research and practice. Building on plot-level assessments of management effects, such as the one carried out in the present research, investigating landscape-scale ES-multifunctionality considering both landscape composition and configuration, while suggesting new or alternative ways to manage grasslands (e.g., increasing plant diversity of swards ¹²), could have the potential to additionally benefit the portfolio of ES provided by agricultural landscapes. Meanwhile, our plot-level results very clearly suggest that diversifying grassland management where this is currently rather homogeneous across farms and landscapes would be an important and effective first step to increase ES-multifunctionality for sustainable grassland management.” (L 426)
--	---

Minor details:	Author comments
L. 34 it needs to be explained what entails intensification of agricultural management in the context of temperate grasslands, and why is that a threat to ES-multifunctionality	Good point, we have added more explanation to this statement in the introduction: “ However, widespread intensification of agricultural management in the form of increased fertilization as well as more frequent and earlier harvests has become a threat for grassland ES-multifunctionality, by heavily focusing on the provisioning and neglecting the other ES ¹²⁻¹⁴.” (L 34)
fig 5. Legend typo in “active recuperation”	Typo corrected, thanks!

Methods:	Author comments
- the use of MLRR to calculate multifunctionality needs to be better justified, at least provide the reference given in the Sup. Material.	In the originally submitted manuscript, the reasons for using the MLRR have been listed in the Supplement. To emphasize this aspect, as requested by the reviewer, we have now shifted a modified version of this previous paragraph to the data analysis section in the revised manuscript. See L 608 ff.

- Also, could you provide a reference for the standardization method used? In principle, the standardization based on normalization that scales the data between 0 and 1 involves: $X \text{ scaled} = (X - X_{\min}) / (X_{\max} - X_{\min})$

Two references to the standardization method are now given (L 555). While standardization from 0 to 1 as described by the reviewer is often used, scaling by the maximum value is also very widely used, and Gamfeldt et al (2017) discusses both methods. There are two major advantages of scaling by the maximum. First, it preserves the mean-to-variance ratio of variables, which is relevant for interpretation, and second, it preserves the effect size of variables in response to a treatment (in our study: the management aspects). Both of these properties are not given by scaling with $(X - X_{\min}) / (X_{\max} - X_{\min})$, and basically, there is no reason why all variables must range down to zero. To exemplify the second case, content variables such as nitrogen content in plant tissue show typically only a small range; for nitrogen content the range is typically 2-4%. While this small range is preserved by scaling with the max, scaling by $(X - X_{\min}) / (X_{\max} - X_{\min})$ “blows” the range considerably up. This is not only misleading regarding the response of such a transformed variable to a treatment, the effect of a variable transformed in this way on a multifunctionality index is also over-weighted. If the response of a variable to a treatment is small on the original scale, it should remain so after transformation as far as possible. Thus, we strongly prefer scaling variables by their maximum, and did so by reasoned arguments.

References used in these answers to the Reviewers:

Allan, E., Manning, P., Alt, F., Binkenstein, J., Blaser, S., Blüthgen, N., ... & Fischer, M. (2015). Land use intensification alters ecosystem multifunctionality via loss of biodiversity and changes to functional composition. *Ecology letters*, 18(8), 834-843.

Gamfeldt, L., & Roger, F. (2017). Revisiting the biodiversity–ecosystem multifunctionality relationship. *Nat Ecol Evol* 1: 0168.

Garland, G., Banerjee, S., Edlinger, A., Miranda Oliveira, E., Herzog, C., Wittwer, R., ... & van Der Heijden, M. G. (2021). A closer look at the functions behind ecosystem multifunctionality: A review. *Journal of Ecology*, 109(2), 600-613.

Hector, A., & Bagchi, R. (2007). Biodiversity and ecosystem multifunctionality. *Nature*, 448(7150), 188-190.

Gamfeldt, L., Roger, F. Revisiting the biodiversity–ecosystem multifunctionality relationship. *Nat Ecol Evol* 1, 0168 (2017). <https://doi.org/10.1038/s41559-017-0168>

Le Provost, G., Schenk, N. V., Penone, C., Thiele, J., Westphal, C., Allan, E., ... & Manning, P. (2023). The supply of multiple ecosystem services requires biodiversity across spatial scales. *Nature ecology & evolution*, 7(2), 236-249.

Maestre, F. T., Quero, J. L., Gotelli, N. J., Escudero, A., Ochoa, V., Delgado-Baquerizo, M., ... & Zaady, E. (2012). Plant species richness and ecosystem multifunctionality in global drylands. *Science*, 335(6065), 214-218.

Manning, P., Van Der Plas, F., Soliveres, S., Allan, E., Maestre, F. T., Mace, G., ... & Fischer, M. (2018). Redefining ecosystem multifunctionality. *Nature ecology & evolution*, 2(3), 427-436.

Soliveres, S., Van Der Plas, F., Manning, P., Prati, D., Gossner, M. M., Renner, S. C., ... & Allan, E. (2016). Biodiversity at multiple trophic levels is needed for ecosystem multifunctionality. *Nature*, 536(7617), 456-459.

Valencia, E., Maestre, F. T., Le Bagousse-Pinguet, Y., Quero, J. L., Tammé, R., Börger, L., ... & Gross, N. (2015). Functional diversity enhances the resistance of ecosystem multifunctionality to aridity in Mediterranean drylands. *New Phytologist*, 206(2), 660-671.

van Der Plas, F., Manning, P., Allan, E., Scherer-Lorenzen, M., Verheyen, K., Wirth, C., ... & Fischer, M. (2016). Jack-of-all-trades effects drive biodiversity–ecosystem multifunctionality relationships in European forests. *Nature communications*, 7(1), 11109.

Van der Plas, F., Ratcliffe, S., Ruiz-Benito, P., Scherer-Lorenzen, M., Verheyen, K., Wirth, C., ... & Allan, E. (2018). Continental mapping of forest ecosystem functions reveals a high but unrealised potential for forest multifunctionality. *Ecology letters*, 21(1), 31-42.

Reviewers' Comments:

Reviewer #1:

Remarks to the Author:

I reviewed the original version of the manuscript as well as the revision. Overall, the authors have done a thorough job in addressing and responding to all my comments raised. In particular, I appreciate that the authors have (i) modified the framing so that it better highlights the actual novelty of this work in the context of the existing literature; (ii) refined the statistical analyses and model specifications by incorporating all the suggestions; and (iii) further improved results presentations and strengthened the discussions.

I have a couple of additional suggestions:

(1) The discussion is a bit lengthy (e.g., 5 single-spaced pages, and 13 paragraphs), which is understandable given the complexity of the analyses and need to address all the comments. The authors may consider consolidating and condensing the discussion.

(2) In Methods section, may consider moving Lines 577-585 out of 'Data analysis'; for example, putting them in the 'Study area, local management practices, and sites' (since these environmental variables were most descriptions of plots within) or in the SI.

(3) Lines 588: It would be helpful to include your justification in the response letter to the text – i.e., topography could influence farmers' decision on how to manage grasslands.

Reviewer #2:

Remarks to the Author:

I appreciate the effort that the authors made to address my earlier comments on the previous version of the manuscript. Overall, the manuscript has improved substantially. In particular, several issues that I raised earlier have been clarified. However, I still have a few suggestions for further improvements:

1. Please clearly define the plot scale and the landscape scale when they first occur in the manuscript. Provide some more explicit explanation of how ES multifunctionality at the two scales is related? How does your "plot-scale multifunctionality" compare and contrast with the term "ecosystem multifunctionality" which is widely used in the literature?

2. Is plot-scale or landscape-scale multifunctionality always preferable? Probably not. Multifunctionality can lead to high degrees of fragmentation, for example. It is landscape sustainability, not landscape multifunctionality, that ought to be the ultimate goal of grassland (or any land) management. Not? Thus, I think that the importance of the study would be elevated to the next level by making the discussion (and conclusions) directly relevant to landscape sustainability.

3. Because grassland management strategies in the world's major grasslands (e.g., prairies in North America, steppes in Eurasia, pampas in SA, and savannas in Africa) are quite different, the title and conclusions should be specific to the study region or similar areas in Europe. It would be interesting and important to briefly discuss the relevance of this study to the major grassland regions around the world.

Reviewer #3:

Remarks to the Author:

Dear Editor,

Authors have done a great job addressing all reviewers' concerns. I noted some additional minor comments to consider before publication.

In general: unify terminology: non-eco-scheme / no Eco-scheme

Abstract. The last sentence of the abstract could provide more concrete information, e.g. incorporating the information from the last sentence of the main text (e.g. replacing "different options exist ..." by "diversifying grassland management ...")

l. 37 correct as: focusing on provisioning ES and neglecting other ES.

l. 45 replace services by ES

l. 46 add a comma (,) before etc.

l. 55 a word is missing in the sentence ("are instruments"?)

l. 150: revise the number of indicators included, in several parts of the text is noted 18 indicators while in l. 147 it is corrected to 17 indicators

l. 274. This sentence is unclear "was mediated by restricting land-use intensity", do you mean was mediated by distinct aspects of land-use intensity? or that the impact could be reduced by restricting land-use intensity?

l. 323. Please rephrase this sentence as the trade-off identified, although very common, is not found everywhere (hence not unavoidable nor obvious). On the other hand, authors do not comment in the discussion that their finding that biomass yield and digestibility were higher in non-eco-scheme plots (l. 117) can be expected (obvious?), as this is how grassland species are selected for.

Reviewer #1 (Remarks to the Author):

I reviewed the original version of the manuscript as well as the revision. Overall, the authors have done a thorough job in addressing and responding to all my comments raised. In particular, I appreciate that the authors have (i) modified the framing so that it better highlights the actual novelty of this work in the context of the existing literature; (ii) refined the statistical analyses and model specifications by incorporating all the suggestions; and (iii) further improved results presentations and strengthened the discussions.

RESPONSE: Thank you very much for the positive feedback!

I have a couple of additional suggestions:

(1) The discussion is a bit lengthy (e.g., 5 single-spaced pages, and 13 paragraphs), which is understandable given the complexity of the analyses and need to address all the comments. The authors may consider consolidating and condensing the discussion.

RESPONSE: Actually, the discussion is 1.5-spaced, which makes it finally shorter than mentioned in the comment. Yet, we checked for unnecessary text and shortened where possible. However, we did this very carefully as we did not want to change too much of the text all reviewers had already agreed on.

(2) In Methods section, may consider moving Lines 577-585 out of 'Data analysis'; for example, putting them in the 'Study area, local management practices, and sites' (since these environmental variables were most descriptions of plots within) or in the SI.

RESPONSE: Changed as suggested. We moved the respective text to the methods section 'Study area, local management practices, and sites' (lines 446ff).

(3) Lines 588: It would be helpful to include your justification in the response letter to the text – i.e., topography could influence farmers' decision on how to manage grasslands.

RESPONSE: Done. To the respective paragraph, we have added: "This was done because topography can influence farmers' decision on how to manage the grassland." (lines 571f).

Reviewer #2 (Remarks to the Author):

I appreciate the effort that the authors made to address my earlier comments on the previous

version of the manuscript. Overall, the manuscript has improved substantially. In particular, several issues that I raised earlier have been clarified. However, I still have a few suggestions for further improvements:

RESPONSE: Thank you for the positive feedback. Please find our detailed response to your remaining comments in the following.

1. Please clearly define the plot scale and the landscape scale when they first occur in the manuscript. Provide some more explicit explanation of how ES multifunctionality at the two scales is related? How does your “plot-scale multifunctionality” compare and contrast with the term “ecosystem multifunctionality” which is widely used in the literature?

RESPONSE: We explain plot- versus landscape-scale multifunctionality quite early in the manuscript (as from **line 61**); there was no suitable earlier spot to bring the definitions. However, we added “ecosystem service multifunctionality” to explain that this is basically the same as plot-scale multifunctionality (**lines 62f**).

2. Is plot-scale or landscape-scale multifunctionality always preferable? Probably not. Multifunctionality can lead to high degrees of fragmentation, for example. It is landscape sustainability, not landscape multifunctionality, that ought to be the ultimate goal of grassland (or any land) management. Not? Thus, I think that the importance of the study would be elevated to the next level by making the discussion (and conclusions) directly relevant to landscape sustainability.

RESPONSE: As we received the recommendation to shorten the discussion, instead of extending it, we would rather not add more to the paper if not very closely related to the content presented. In this regard, the concept of landscape sustainability would certainly require longer elaborations to introduce and also explain the term/concept, which is quite fuzzy and – to our knowledge - not very well established.

3. Because grassland management strategies in the world’s major grasslands (e.g., prairies in North America, steppes in Eurasia, pampas in SA, and savannas in Africa) are quite different, the title and conclusions should be specific to the study region or similar areas in Europe. It would be interesting and important to briefly discuss the relevance of this study to the major grassland regions around the world.

RESPONSE: We added a short sentence to the discussion (last paragraph on the future outlook, **lines 400ff**) specifying that it would be good to assess the effects of management aspects specifically also for natural grasslands and its particular management.

Reviewer #3 (Remarks to the Author):

Dear Editor,

Authors have done a great job addressing all reviewers’ concerns. I noted some additional minor comments to consider before publication.

RESPONSE: Thank you for the positive feedback. Please find our detailed response to your remaining comments in the following.

In general: unify terminology: non-eco-scheme / no Eco-scheme

RESPONSE: Was changed as suggested.

Abstract. The last sentence of the abstract could provide more concrete information, e.g. incorporating the information from the last sentence of the main text (e.g. replacing “different options exist ...” by “diversifying grassland management ...”)

RESPONSE: Was changed as suggested.

I. 37 correct as: focusing on provisioning ES and neglecting other ES.

RESPONSE: Was changed as suggested.

I. 45 replace services by ES

RESPONSE: Was changed as suggested.

I. 46 add a comma (,) before etc.

RESPONSE: Was changed as suggested.

I. 55 a word is missing in the sentence (“are instruments”?)

RESPONSE: Indeed, “are” was missing. Thanks for this hint.

I. 150: revise the number of indicators included, in several parts of the text is noted 18

indicators while in I. 147 it is corrected to 17 indicators

RESPONSE: Was corrected to 17 ES-indicators throughout the manuscript. Figure 4 was changed accordingly.

I. 274. This sentence is unclear “was mediated by restricting land-use intensity”, do you mean was mediated by distinct aspects of land-use intensity? or that the impact could be reduced by restricting land-use intensity?

RESPONSE: The sentence was rephrased to clarify and now reads: “The impact of the three management aspects on plot-scale ES-multifunctionality was mainly related to lowering land-use intensity.” (Lines 248f)

I. 323. Please rephrase this sentence as the trade-off identified, although very common, is not found everywhere (hence not unavoidable nor obvious). On the other hand, authors do not comment in the discussion that their finding that biomass yield and digestibility were higher in non-eco-scheme plots (I. 117) can be expected (obvious?), as this is how grassland species are selected for.

RESPONSE: We rephrased and deleted “obviously unavoidable” from the sentence (line 295ff). Regarding the finding that biomass yield and digestibility were higher in non-eco-scheme plots, this was indeed expected and is in line with previous work. It was, however, necessary to include these provisioning ES to show the effects of management on the whole spectrum of ES-multifunctionality. We highlight the effect of Eco-scheme on provisioning ES now in line 239. We decided against mentioning this effect was expected as this would unnecessarily extend the manuscript, and also because we do not discuss this (obvious) effects in much detail in the paper.